# CymbaDiff: Structured Spatial Diffusion for Sketch-based 3D Semantic Urban Scene Generation

Li Liang[1]   Bo Miao[2]   Xinyu Wang[1]   Naveed Akhtar[3]   Jordan Vice[1]   Ajmal Mian[1]

[1] The University of Western Australia
[2] AIML, The University of Adelaide
[3] The University of Melbourne

## Abstract

Outdoor 3D semantic scene generation produces realistic and semantically rich environments for applications such as urban simulation and autonomous driving. However, advances in this direction are constrained by the absence of publicly available, well-annotated datasets. We introduce SketchSem3D, the first large-scale benchmark for generating 3D outdoor semantic scenes from abstract freehand sketches and pseudo-labeled annotations of satellite images. SketchSem3D includes two subsets, Sketch-based SemanticKITTI and Sketch-based KITTI-360 (containing LiDAR voxels along with their corresponding sketches and annotated satellite images), to enable standardized, rigorous, and diverse evaluations. We also propose Cylinder Mamba Diffusion (CymbaDiff) that significantly enhances spatial coherence in outdoor 3D scene generation. CymbaDiff imposes structured spatial ordering, explicitly captures cylindrical continuity and vertical hierarchy, and preserves both physical neighborhood relationships and global context within the generated scenes. Extensive experiments on SketchSem3D demonstrate that CymbaDiff achieves superior semantic consistency, spatial realism, and cross-dataset generalization. The code and dataset will be available at https://github.com/Lillian-research-hub/CymbaDiff.

## 1   Introduction

Generative modeling has demonstrated remarkable progress in the 2D and 3D domains, largely fueled by the rapid development of diffusion models [1, 2, 3, 4]. In 3D, diffusion approaches have significantly advanced 3D object synthesis [5, 6] and indoor scene generation [7, 8]. However, generating large-scale 3D outdoor environments remains widely underexplored [9, 10, 11], as outdoor urban scenes pose greater challenges due to their higher semantic diversity, complex spatial structures, and dynamic contextual dependencies. Despite these challenges, synthesizing realistic and scalable 3D urban scenes is increasingly critical, as it underpins a wide range of emerging applications, including city-scale simulation [12, 13] and autonomous driving [14, 15, 16, 17].

A few methods have recently surfaced for 3D outdoor scene generation [9, 10, 11, 18, 19], often relying on bird's-eye view (BEV) with only road data or multi-scale scene hierarchies to guide generation. BEV-based approaches suffer from insufficient 3D structural information, limiting both semantic richness and geometric fidelity. Meanwhile, modeling multi-scale scene hierarchies typically requires generative models to repeatedly synthesize scenes at multiple spatial resolutions, increasing both computational and structural complexity. Moreover, due to the lack of a public large-scale benchmark, current approaches typically use self-curated and heavily preprocessed datasets for evaluation [9], which fundamentally constrains rigorous benchmarking. Sketch-based methods [20, 21, 22, 23] have recently emerged as a promising paradigm for user-guided 3D generation, enabling intuitive control through freehand drawings. However, their applicability remains confined to the synthesis of isolated 3D objects or simple indoor scenes. Expanding sketch-based 3D reconstruction to outdoor scenes is currently widely open. Challenges in this novel pursuit arise from complex

scene layouts, diverse object geometries, and the need to preserve spatio-semantic coherence across large-scale scenes.

This work takes a significant step towards extending sketch-based generation to outdoor environments. To that end, we build upon the growth of State Space Models (SSMs) [24], which have gained increased attention across image segmentation [25, 26] and point cloud processing [27, 28] for their ability to capture long-range dependencies while remaining efficient through selective computation.However, to enhance global contextual understanding, SSMs typically aggregate information from multiple scan directions, leading to substantial memory overhead. Moreover, the scanning order imposed by the Cartesian coordinate system can distort local neighborhood relationships, especially in scenes with limited spatial coherence.

To address the above-noted challenges for sketch-based 3D outdoor scene generation, we first present 'SketchSem3D', a large-scale dataset tailored for the task. SketchSem3D enables the synthesis of semantically rich outdoor 3D environments from freehand sketches and pseudo-labeled satellite image annotations. The annotation pipeline properly integrates CLIP-based textual guidance [29] with image embeddings from the Segment Anything Model (SAM) [30], enabling robust and automated semantic labeling. SketchSem3D comprises two subsets, Sketch-based SemanticKITTI and Sketch-based KITTI-360, designed to support standardized benchmarking and fair comparison. Building upon this dataset, we define the novel 'sketch-based 3D outdoor scene generation' research task. We also propose Cylinder Mamba Diffusion, the first approach to handle this task. As adjacent Cartesian-based voxel sequences may misrepresent spatial proximity in outdoor scenes, CymbaDiff is particularly tailored to handle voxel discrepancies. Our underlying model is a denoising network, combining an SSM architecture with generative diffusion in the latent space. We design cylinder mamba blocks to enhance spatial coherence during the generative process, imposing a structured spatial ordering to explicitly encode cylindrical continuity and vertical hierarchy, preserving spatial neighborhood relationships within scenes.

Our key contributions are summarized below:

- We introduce the novel task of 'sketch-based 3D outdoor scene generation', which enables intuitive and flexible user interaction through freehand sketches and pseudo-labeled satellite image annotations. By reducing the need for manual semantic annotation, this task offers an efficient solution to generate training data for applications such as urban-scale simulation and autonomous driving.

- We present SketchSem3D, the first public large-scale sketch-based benchmark for 3D outdoor semantic scene generation. It includes two subsets, Sketch-based SemanticKITTI and Sketch-based KITTI-360, and enables standardized benchmarking for the development and evaluation of generative models in complex outdoor settings.

- We propose CymbaDiff, a generative model that incorporates the proposed cylinder mamba blocks to enhance spatial coherence during the generation process. We also conduct extensive experiments on the Sketch-based SemanticKITTI and Sketch-based KITTI-360 benchmarks, demonstrating state-of-the-art performance in 3D semantic scene generation and completion.

## 2 Related Work

### 2.1 State Space Models

Recent studies have demonstrated the strong capability of State-Space Models (SSMs) in capturing long-range dependencies across sequential data [31, 32]. These models have been successfully applied in a variety of domains, including medical image segmentation [25, 33], image restoration [34, 35], natural language processing (NLP) [36, 37], and point cloud processing [38, 28]. Many of these approaches build upon foundational architectures such as VisionMamba [39], S4ND [40], and Mamba-ND [41]. Specifically, VisionMamba [39] integrates bidirectional SSMs for data-dependent global context modeling and employs positional embeddings to enhance location-aware visual recognition. S4ND [40] extends the SSM framework by incorporating local convolution operations, thereby enabling processing beyond one-dimensional inputs. Mamba-ND [41] further addresses multi-dimensional data by utilizing various scan patterns within a single block to enhance performance in discriminative tasks. Despite their strengths, these methods primarily focus on maximizing contextual information through multiple scanning directions, often neglecting structured spatial coherence across horizontal and vertical hierarchies, particularly under memory-constrained settings.

## 2.2 3D Semantic Scene Generation

Diffusion models have evolved from generating 2D images to addressing increasingly complex 3D data modeling tasks [2]. Compared to traditional generative models such as Generative Adversarial Networks (GANs) [42] and Variational Autoencoders [43], diffusion models follow a progressive denoising process [44], which enhances training stability and improves the capacity to capture complex data distributions. These advantages render diffusion models particularly suitable for 3D data generation tasks. While much of the existing research has focused on object-level synthesis [45, 46, 47, 48, 49, 50, 51, 52] and indoor scene generation [53, 54, 55, 56], there is a growing body of work exploring 3D outdoor semantic scene generation [57, 11, 10, 58, 47, 9] as it underpins a wide range of emerging applications, including autonomous driving [14, 15, 16, 17] and city-scale simulation [12, 13]. For instance, UrbanDiff [9] conditions generation on BEV maps to produce urban scenes in the form of semantic occupancy grids, integrating both geometry and semantic information. P-DiscreteDif [10] proposes a progressive multi-scale strategy that synthesizes large-scale 3D scenes by conditioning each stage on the output from the preceding resolution level, with the initial model conditioned solely on noise. Despite these advancements, the absence of standardized datasets for 3D outdoor semantic scene generation has led to the use of heterogeneous benchmarks with inconsistent scene conditions, thereby limiting fair comparison and hindering systematic progress in the field.

## 2.3 3D Semantic Scene Completion

3D semantic scene completion methods can be broadly categorized into four categories: image-based approaches [59, 60, 61], point cloud-based methods [62, 63], voxel-based techniques [64, 65], and multi-modality-based frameworks [66, 67]. Most existing methods are built upon convolutional neural networks (CNNs) or Transformer-based architectures. For instance, Xia *et al.* [65] propose a CNN network (SCPNet), which enhances single-frame scene completion by incorporating dense relational semantic knowledge distillation along with a label rectification strategy to mitigate artifacts introduced by dynamic objects. CGFormer [59] enhances semantic scene completion by introducing a context- and geometry-aware voxel transformer, which initializes queries based on the contextual information from individual input images and extends deformable cross-attention mechanisms from 2D image space to 3D voxel space. While CNNs are computationally efficient, they are inherently limited by their receptive field size. Transformers address this limitation by enabling global context modeling but come with high memory costs. Recently, Segmamba [25] has emerged as a promising alternative, offering a favorable trade-off by supporting large receptive fields with improved memory efficiency, making it suitable for 3D semantic scene completion.

# 3 SketchSem3D Dataset

Sketch-based methods have recently gained increasing attention as a promising paradigm for user-guided 3D modeling, offering intuitive and flexible interaction through freehand drawing. While these approaches show great potential, they are constrained to generating isolated 3D objects and lack the capacity to model complex, semantically rich scenes. In a related direction, UrbanDiff [9] introduced BEV representations as conditional inputs for 3D semantic scene generation. By leveraging the spatial alignment between 2D projections and 3D structures, this approach promotes 2D-to-3D consistency. However, BEV-based supervision inherently constrains the diversity of the generated scenes. Moreover, acquiring BEV images that accurately reflect the semantic layout of complex 3D environments is particularly challenging in outdoor settings.

We propose a sketch-based framework for 3D outdoor semantic scene generation. It enables users to define scene layouts using coarse freehand sketches combined with pseudo-labeled satellite image annotations, facilitating a more natural and accessible interaction modality. By circumventing the need for labor-intensive annotations and large-scale sensor-based data collection, the framework significantly enhances scalability. We leverage this framework in the design of our SketchSem3D benchmark dataset.

## 3.1 Benchmark Construction

The benchmark comprises two distinct datasets, Sketch-based SemanticKITTI and Sketch-based KITTI-360, each constructed through a systematic three-stage pipeline discussed below.

**Data Sourcing.** We construct the two datasets using the 3D ground truth (GT) from SemanticKITTI [68] and SSCBench-KITTI-360 [69], respectively. Each scene is enriched with freehand sketches and pseudo-labeled satellite image annotations to enable conditioned 3D scene generation. Both datasets comprise five components: freehand (like) sketches, satellite images, pseudo-labeled

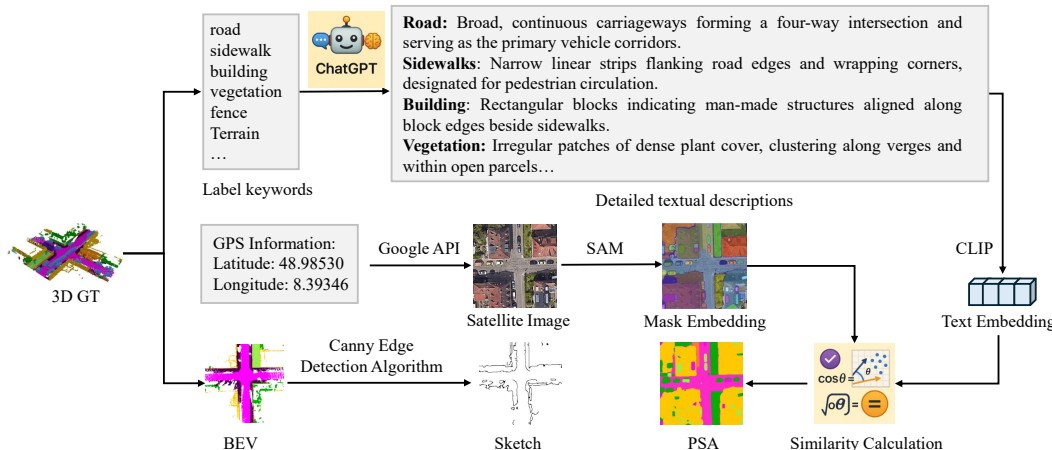

Figure 1: Pipeline for SketchSem3D construction. SAM and PSA denote Segment Anything Model and Pseudo-labeled Satellite Image Annotations, respectively.

Table 1: Comparing SketchSem3D (last two rows) with BEV-based NuScenes.

| Dataset | Pairs | Condition | 3D Geospatial Semantics | Classes | 3D GT Voxels |
|---------|-------|-----------|-------------------------|---------|--------------|
| BEV-based NuScenes [9] | 34149 | BEV | ✗ | 17 | $192 \times 192 \times 16$ |
| Sketch-based SemanticKITTI | 58987 | Sketch / PSA | ✓ | 20 | $256 \times 256 \times 32$ |
| Sketch-based KITTI-360 | 36057 | Sketch / PSA | ✓ | 19 | $256 \times 256 \times 32$ |

annotations, semantic label keywords, and 3D GT (output). Figure 1 shows the dataset construction pipeline. The 3D GT is extended from the respective source datasets. Sketches are generated by applying the Canny edge detector [70] to BEV projections of 3D GT. These sketches closely resemble freehand drawings, which can be more easily produced at test time compared to BEV projections, providing abstract representations of scene geometry.

Semantic categories (e.g., road, tree, vehicle) are also available as GT and recorded as label keywords without spatial encoding. To enrich the semantic context, GPT-4 [71] is used to generate descriptive texts for each category, supporting alignment with visual features. We leverage the GPS information provided in KITTI [72] and KITTI-360 [73] to retrieve the corresponding satellite images. We then apply CLIP [29] to encode the enriched contextual descriptions and SAM [30] to obtain mask-level embeddings from the satellite images. By computing the cosine similarity between text and image embeddings, we infer the semantic composition of each scene from the satellite perspective, producing the pseudo-labeled annotations used in our SketchSem3D dataset.

**Data Filtering and Formatting.** To address any semantic labeling errors or inconsistencies in the automated alignment between CLIP [29] text embeddings and SAM [30] image mask embeddings, we perform a *manual review* of the resulting class distributions to ensure annotation accuracy and dataset reliability. Each sketch-based dataset consists of five components: (*i*) the sketch, (*ii*) satellite image, (*iii*) pseudo-labeled satellite image annotations, (*iv*) label keywords, and (*v*) 3D GT. The sketch is stored as a binary edge map in image format, capturing the structural outline of the scene. The satellite image is a geo-referenced RGB image of the same size, spatially aligned with the GPS coordinates of the corresponding scene. The pseudo-labeled satellite image annotations are single-channel semantic maps, where each pixel represents a semantic class ID. Although two-dimensional, these annotations provide coarse semantic cues that serve as important conditional guidance for reconstructing 3D voxel scenes. The label keywords for each scene are saved in a .txt file indexed by scene ID, listing the semantic class keywords present in the scene. Finally, 3D GT is provided as a volumetric label map, where each voxel is assigned a semantic class encoded as a 16-bit unsigned integer, following the format of SemanticKITTI [68].

## 3.2 Data Statistics Comparison and Evaluation Metrics

Table 1 compares our SketchSem3D dataset with the BEV-based NuScenes dataset [9]. We can see that our dataset is better in every aspect offering higher resolution, more classes, additional geospatial

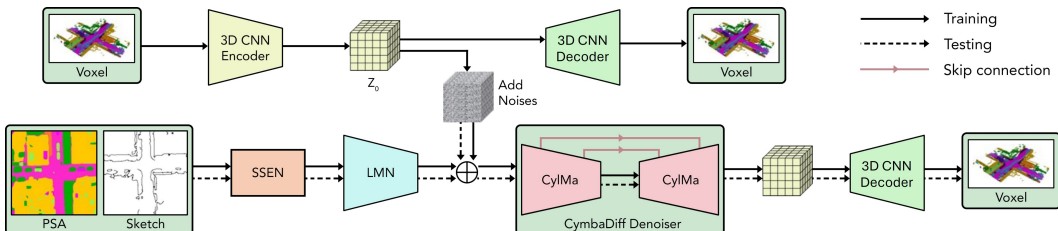

Figure 2: Architecture of our CymbaDiff generation network. The Scene Structure Estimation Network (SSEN) extracts abstract structural information from Pseudo-labeled Satellite Image Annotations (PSA) and the Sketch. The Latent Mapping Network (LMN) compresses the input conditions into a latent representation, which is then processed by the CymbaDiff denoiser, which utilizes the proposed cylinder mamba blocks (CylMa) to perform latent denoising.

semantics, two conditions instead of one and contains a much larger number of 3D scenes (total 95,044 compared to 34,149 in [9]). Notably, each subset of SketchSem3D contains more frames than [9]. Moreover, our conditions (sketch and PSA) are easier to obtain at test time, enhancing the practicality. Sketch-based SemanticKITTI includes 58,172 training and 815 validation frames, while Sketch-based KITTI-360 consists of 33,892 training and 2,165 validation frames. In SketchSem3D, all satellite images, sketches, and pseudo-labeled annotations are standardized to a resolution of $256 \times 256$ pixels, with corresponding 3D GT of $256 \times 256 \times 32$ voxels. In comparison, BEV-based NuScenes [9] contains 3D GT of $192 \times 192 \times 16$ voxels and lacks explicit geospatial structure as well as detailed 3D semantic distribution.

To evaluate the quality and diversity of the generated 3D semantic scenes, we adopt two widely used metrics: Fréchet Inception Distance (FID) [74] and Maximum Mean Discrepancy (MMD) [9]. Together, these metrics capture statistical similarity and feature-level realism, providing a comprehensive assessment of generative performance. Further details on the evaluation metrics are supplied in the Appendix.

## 4 Method

We propose a 3D semantic scene generation method that captures both geometric structure and semantic information, based on a given sketch and its corresponding pseudo-labeled satellite image annotations. Formally, let the sketch image be denoted as $I \in \mathbb{R}^{L \times W \times 1}$, and the associated pseudo-labeled satellite image annotations as $PSA \in \mathbb{R}^{L \times W \times 1}$. These two modalities are jointly projected into a structured 3D voxel grid $\mathbb{R}^{L \times W \times H \times 1}$, which encodes the spatial structure of the semantic scene, where $L, W, H$ represent the length, width, and height of the 3D space, respectively. The goal is to generate a semantically complete 3D scene by predicting each voxel's occupancy state and semantic label. Each voxel in the generated grid is assigned a semantic class label $c \in 0, 1, 2, \ldots, C - 1$, where $C$ is the total number of semantic categories. By convention, $c = 0$ corresponds to empty or unoccupied space, while the remaining values represent distinct semantic classes.

### 4.1 Scene Structure Estimation Network

To facilitate efficient convergence of CymbaDiff, we introduce a scene structure estimation network (SSEN) that produces a coarse structural representation of the target 3D scene, as shown in Figure 2. This structural prior guides the diffusion model towards geometrically plausible outputs during early generation steps. Inspired by recent advances in structural scene modeling [65, 75], the SSEN architecture incorporates multi-scale feature extraction modules with Dimensional Decomposition Residual (DDR) blocks. Specifically, multi-scale feature extraction modules capture hierarchical contextual information by aggregating features across multiple receptive fields. It employs parallel branches of $3 \times 3 \times 3$ convolutions to replace $5 \times 5 \times 5$ and $7 \times 7 \times 7$ convolutions, which are progressively stacked and merged at multiple levels, as shown in Figure 3 (b). The DDR structure decomposes a standard $k \times k \times k$ 3D convolution into a sequence of three separable layers: $1 \times 1 \times k$, $1 \times k \times 1$, and $k \times 1 \times 1$, as illustrated in Figure 3 (d). The multi-scale modules capture spatial context and semantically-rich features across different receptive fields, while the DDR blocks enhance the network's representational capacity with limited computational cost. Through joint use of these components, SSEN generates a voxel-based structural representation that accelerates convergence during the diffusion-driven 3D generation while improving geometric fidelity.

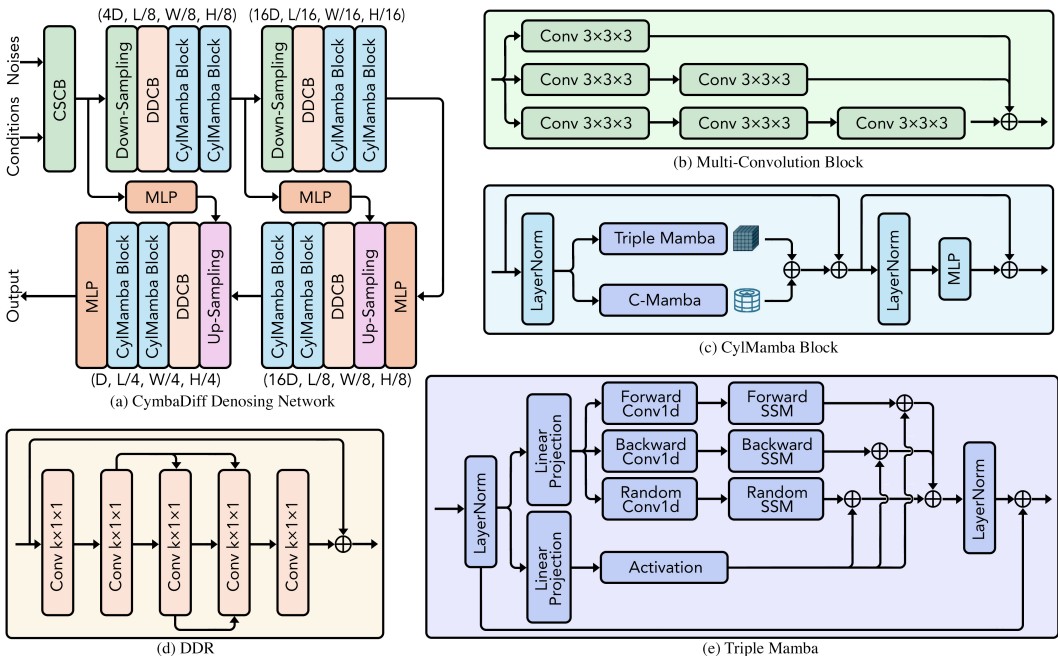

Figure 3: Architecture of the CymbaDiff denoising network. CylMamba denotes cylinder mamba block. Refer to the text for details.

## 4.2 Variational Autoencoder (VAE) / Latent Mapping Network

As illustrated in Figure 2, CymbaDiff operates in the latent space of a VAE, which provides a compact and informative representation for 3D semantic scenes. The VAE is trained with a combination of cross-entropy loss [76] and Lovász-Softmax loss [77]. This joint objective encourages alignment with the voxel grid manifold while mitigating the blurriness often introduced by conventional voxel-wise losses (like $L_2$ [78]). Given a voxelized input scene $V \in L \times W \times H$, the encoder $\mathbb{E}$ maps it to a latent representation $z = \mathbb{E}(V)$, the decoder $\mathbb{D}$ then reconstructs the scene as $\tilde{V} = \mathbb{D}(z) = \mathbb{D}(\mathbb{E}(V))$. In our implementation, $\mathbb{E}(\cdot)$ reduces the spatial resolution of the input voxel grid by a factor of $f = 4$, effectively compressing the scene while preserving key structural features. The VAE encoder consists of two down-sampling blocks, each comprising four consecutive convolutional layers. Every pair of convolutional layers is followed by a Batch Normalization layer and a ReLU activation function. Following these operations, a downsampling convolutional layer is applied, which is also followed by Batch Normalization and ReLU. To align with the VAE's latent distribution, the latent mapping network is designed to share the same architecture as the encoder.

## 4.3 Cross-Scale Contextual Block / Dilated Decomposed Convolution Block

We introduce the Cross-Scale Contextual Block (CSCB), inspired by hierarchical receptive fields in VGG [79] and multi-path processing in SCPNet [65]. CSCB efficiently captures local-to-global context from conditioning inputs with minimal memory overhead. Starting with a $3 \times 3 \times 3$ convolution, it has cascaded multi-covolution blocks (see Figure 3 (b)) with skip connections, and ends with another $3 \times 3 \times 3$ convolution before adding the residual output. Moreover, the Dilated Decomposed Convolution Block (DDCB) employs DDR blocks [75] with varying dilation rates of 1, 2 and 3 to capture diverse contextual features. The DDR structure is shown in Figure 3(d). The DDR block reduces computational cost of $C^{in} \times C^{out} \times k^3$ in traditional 3D convolutions to $C^{in} \times C^{out} \times 3k$ by breaking down the operations into $1 \times 1 \times k$, $1 \times k \times 1$, and $k \times 1 \times 1$ layers, which decreases the parameter count three times while maintaining detailed spatial layout. Therefore, this decomposition significantly reduces the number of parameters while preserving fine-grained spatial layout information.

## 4.4 CymbaDiff Denoising Network

As shown in Figure 2 (a), CymbaDiff generates scenes from conditional inputs and latent noise, drawing on the Mamba framework [80] to model sequences through a state-space formulation. A continuous input $x(t) \in \mathbb{R}$ is transformed into an output $y(t) \in \mathbb{R}$ via an intermediate hidden state

$h(t) \in \mathbb{R}^N$, before being discretized. The SSMs model [81] is typically formulated using linear ordinary differential equations (ODEs), defined as:

$$h^{'}(t) = Ah(t) + Bx(t), \quad y(t) = Ch(t), \tag{1}$$

where $A \in \mathbb{R}^{N \times N}$ and $B \in \mathbb{R}^{N \times 1}$, $C \in \mathbb{R}^{1 \times N}$ denote the state matrix, input matrix, and output matrix, respectively. Since deriving the analytical solution for $h(t)$ is often intractable and real-world data is typically discrete, the system is discretized as follows:

$$h(t) = \overline{A}h(t-1) + \overline{B}x(t), \quad y(t) = \overline{C}h(t), \tag{2}$$

where $\overline{A} = \exp(\triangle A)$ and $\overline{B} = (\triangle A)^{-1}(\exp(\triangle A) - I) \cdot \triangle B$, $\overline{C} = C$ are the discretized state parameters and $\triangle$ is the discretization step size. The final output is obtained by applying a global convolution over a structured kernel. The downsampling and upsampling operations follow the design proposed in [25].

**Cylinder Mamba Block.** A core component of the CymbaDiff denoiser is the cylinder mamba block, illustrated in Figure 3 (c). This block integrates the Triple Mamba module [82] with our proposed cylinder mamba layer design to jointly leverage the advantages of both Cartesian and cylindrical coordinate representations. The Triple Mamba module, based on Cartesian grids, effectively preserves precise geometric distances, critical for modeling local physical neighborhoods. However, adjacent elements in Cartesian voxel sequences may misrepresent spatial relationships, limiting the effectiveness of sequential modeling. In contrast, the cylinder mamba layer $(\theta, r, z)$ imposes a structured spatial ordering that explicitly captures cylindrical continuity and vertical hierarchy. This ordering provides a vehicle-centric, geometrically coherent view, enabling angular-radial semantic tokenisation and supporting long-range context modelling with Mamba, for example, capturing structural information about sidewalks and buildings flanking the road.

The detailed structure of Triple Mamba layer is illustrated in Figure 3(e), and the cylinder mamba (C-Mamba) layer adopts the same architecture. Before entering the Mamba layers, input features undergo residual Layer Normalization ($LN$) on respective coordinate-based feature representation i.e, $z_{TMB}(t) = (LN(f_{TMB}(t))) + f_{TMB}(t)$ and $z_{CMB}(t) = (LN(f_{CMB}(t))) + f_{CMB}(t)$. $f_{TMB}(t)$ and $z_{TMB}(t)$ are the input and output features before the Triple Mamba layer, while $f_{CMB}(t)$ and $z_{CMB}(t)$ denote the corresponding features before cylinder mamba layer. The temporal dynamics of the Triple Mamba and C-Mamba layer input are thus governed by:

$$h(t) = \overline{A}h(t-1) + \overline{B}z_{TMB}(t), \quad y(t) = \overline{C}h(t), \tag{3}$$

$$h(t) = \overline{A}h(t-1) + \overline{B}z_{CMB}(t), \quad y(t) = \overline{C}h(t). \tag{4}$$

The Triple Mamba layer and C-mamba layer apply three separate Mamba modules, each operating on the same input $z_{TMB}(t)$ and $z_{CMB}(t)$ but with distinct ordering strategies: forward ($\psi_i^f$), backward ($\psi_i^b$), and random inter-slice ($\psi_i^u$) directions. The output of the $i^{th}$ Triple Mamba layer and C-mamba layer are computed as:

$$\psi_i(z_{TMB}(t)) = \psi_i^f(z_{TMB}(t)) + \psi_i^b(z_{TMB}(t)) + \psi_i^u(z_{TMB}(t)), \tag{5}$$

$$\omega_i(z_{CMB}(t)) = \omega_i^f z_{CMB}(t)) + \psi_i^b(z_{CMB}(t)) + \psi_i^u(z_{CMB}(t)), \tag{6}$$

where $\psi_i(z_{TMB}(t))$ and $\omega_i(z_{CMB}(t))$ represent the outputs of the $i^{\text{th}}$ triple Mamba and C-mamba layer. Fused 3D features from triple Mamba and C-mamba layers are formulated as $\psi_i^{all} = \phi_i^{all}(z_{TMB}(t)) + \omega_i^{all}(z_{CMB}(t))$, where $\phi_i^{all}(z_{TMB}(t)) = \text{MLP}(LN(\psi_i(z_{TMB}(t)))) + \psi_i(z_{TMB}(t))$ and $\omega_i^{all}(z_{CMB}(t)) = \text{MLP}(LN(\omega_i(z_{CMB}(t)))) + \omega_i(z_{CMB}(t))$. $\phi_i^{all}(z_{TMB}(t))$ and $\omega_i^{all}(z_{CMB}(t))$ denote the output feature from the triple Mamba and the C-mamba layer. MLP corresponds to stacked linear layers. Note that the input features in the C-Mamba layer are sorted by angular, radial, and vertical indices ($(\theta, r, z)$), and the output features are mapped back to Cartesian spatial ordering $(x, y, z)$ (the same ordering in the Triple Mamba layer) and fused with those from the Triple Mamba layer, allowing the model to jointly exploit radial and axis-aligned spatial cues. This joint representation enhances the model's ability to learn both local and global 3D spatial structures, capturing both Cartesian and cylindrical representations. Unlike the original Mamba [41, 83], which emphasizes directional context aggregation along scan lines with higher memory usage, our cylinder mamba block is specifically designed to efficiently capture spatially-structured 3D information.

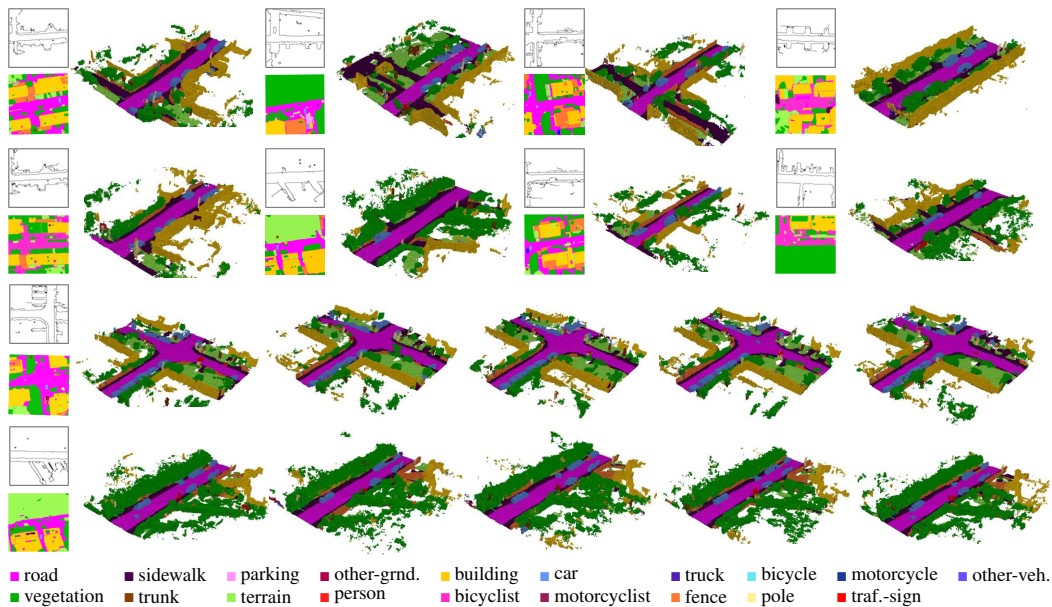

road ■ sidewalk ■ parking ■ other-grnd. ■ building ■ car ■ truck ■ bicycle ■ motorcycle ■ other-veh.
■ vegetation ■ trunk ■ terrain ■ person ■ bicyclist ■ motorcyclist ■ fence ■ pole ■ traf.-sign

Figure 4: Qualitative results on the Sketch-based SemanticKITTI validation set. The 1st and 2nd rows show generated scenes conditioned on the corresponding freehand sketch and pseudo-labeled satellite images. The 3rd and 4th rows demonstrate the model's capability to generate moderately diverse 3D scenes (with different details) under identical input conditions.

## 5 Experiments

**Implementation Details.** Our model is trained on the Sketch-based SemanticKITTI training split from the SketchSem3D dataset. For evaluation, we use the validation splits of both the Sketch-based SemanticKITTI and Sketch-based KITTI-360 subsets, also from SketchSem3D. Following UrbanDiff [9], we train a dedicated network to extract latent features that encode both geometric and semantic information. These features are used to compute 3D FID and MMD, providing a joint assessment of generation quality and distributional similarity to ground-truth scenes. Additional implementation details are presented in the Appendix.

### 5.1 3D Semantic Scene Generation and Ablation Study

**3D Semantic Scene Generation.** As shown in Table 2, we compare our approach with two recent state-of-the-art baselines, SSD [57] and Semcity [11]. Across all evaluation metrics, our method consistently achieves superior performance. Notably, on the Sketch-based SemanticKITTI subset, it improves the FID score by approximately 16 points compared to Semcity [11], highlighting its effectiveness in interpreting sparse and abstract conditional inputs, such as freehand sketches and pseudo-labeled satellite image annotations.

SSD [57] and Semcity [11] both adopt 2D FID for evaluation. In contrast, we adopt more comprehensive 3D evaluation metrics, 3D FID and MMD, that more accurately assess geometric fidelity and semantic consistency in voxel space. To evaluate the effectiveness of our approach, we replace the CymbaDiff denoising network with two baselines: a 3D extension of the Latent Diffusion network [1] and the 3D DiT model [84], and conduct experiments on the SketchSem3D benchmark. Results in Table 2 show that our method consistently outperforms both baselines, demonstrating its superior performance in 3D semantic scene generation. For additional context, UrbanDiff [9] reports competitive performance, with a 3D FID of 291.4 and a 3D MMD of 0.11 on the NuScenes dataset. However, their experimental setting is less challenging, as the voxel resolution of NuScenes is $192 \times 192 \times 16$ and with only 17 semantic classes. In comparison, our benchmark dataset has a resolution of $256 \times 256 \times 256$ and with 20 classes for the Sketch-based SemanticKITTI subset and 16 classes for the Sketch-based KITTI-360 subset. The primary factor underlying this is that UrbanDiff operates solely within the Cartesian coordinate system, leading to the loss of important volumetric structural information. Furthermore, UrbanDiff does not release its source code or preprocessed data, which prevents direct comparison with our proposed task.

Table 2: Semantic scene generation results. SK: sketch, PSA: pseudo-labeled satellite image annotations. SSD and Semcity: 2D FID.

| Datasets | Method | Condition | FID ↓ | MMD ↓ |
|---|---|---|---|---|
| SemanticKITTI | SSD [57] | - | 112.82 | - |
| | Semcity [11] | - | 56.55 | - |
| | 3D Latent Diffusion [1] | SK+PSA | 165.65 | 0.09 |
| | 3D DIT [84] | SK+PSA | 138.86 | 0.08 |
| | CymbaDiff (ours) | SK+PSA | **40.67** | **0.04** |
| KITTI-360 | 3D Latent Diffusion [1] | SK+PSA | 330.86 | 0.12 |
| | 3D DIT [84] | SK+PSA | 272.83 | 0.11 |
| | CymbaDiff (ours) | SK+PSA | **107.53** | **0.08** |

Table 3: Ablation study on Sketch-based SemanticKITTI test set. w/o: "without", C-Mamba: cylinder mamba.

| Method | FID ↓ | MMD ↓ |
|---|---|---|
| w/o CSCB | 90.53 | 0.06 |
| w/o DDCB | 76.57 | 0.06 |
| w/o C-Mamba | 74.09 | 0.05 |
| CymbaDiff | 40.67 | 0.04 |

In Table 2, to evaluate robustness and generalization, we directly applied our model, trained only on Sketch-based SemanticKITTI, to Sketch-based KITTI-360 without any fine-tuning. During this evaluation, only the overlapping class labels (16 classes) between the two subsets are used. Our model maintains top-tier performance, producing structurally coherent and semantically meaningful 3D scenes. This cross-dataset evaluation highlights the strong generalization capability of our approach.

We present qualitative results on the Sketch-based SemanticKITTI validation set in Figure 4. Rows 1 and 2 illustrate the generated semantic scenes conditioned on the input sketches and their corresponding PSAs. Rows 3 and 4 present additional generation results using the same input conditions to demonstrate both consistency and moderate diversity in scene synthesis. We see that our model effectively produces structurally accurate, and semantically meaningful 3D scenes that align well with inputs. These visualizations further demonstrate the model's ability to integrate abstract freehand sketches and pseudo-labeled satellite cues to generate high-quality semantic reconstructions. Some sketch-PSA pairs may have differences because the 3D ground truth annotations in SemanticKITTI were collected around 2013 and the satellite images used for PSA were captured around 2025. PSA generation, being automatic, is also prone to errors. In contrast, sketches originate directly from the 2013 ground-truth data, maintaining temporal consistency and serving as a stable spatial reference to mitigate the domain gap.

Observing the results of CymbaDiff on the proposed SketchSem3D dataset, it is apparent that it demonstrates strong performance, effectively handling challenges such as semantic misalignment caused by noisy pseudo-labels, e.g., due to confusion between vegetation and buildings. Nevertheless, this method does occasionally fail to accurately reconstruct small or occluded objects that are underrepresented in the training data or sparsely encoded in the sketch and PSA inputs. Although CymbaDiff mitigates this issue to some extent through the use of the Cross-Scale Contextual Block and Cylinder Mamba Block, which capture multi-scale contextual information, its performance could be further enhanced by increasing the representation of small objects in the dataset.

**Ablation Study.** We conducted systematic experiments to evaluate the impact of different components in our model and to quantify their individual contributions to the overall performance. As presented in Table 3, the ablation study offers valuable insights into the role and effectiveness of each component. These results allow us to isolate and identify the elements that most significantly enhance the model's performance in the 3D semantic scene generation task. Notably, the CSCB, DDCB, and cylinder mamba blocks play a critical role, as they enable the model to capture complex spatial and semantic relationships within 3D scenes more effectively. "w/o C-Mamba" refers to a variant that retains only the triple Mamba layers.

## 5.2 3D Semantic Scene Completion.

Since our work explores a new research direction and, currently, there are no directly comparable methods using the same input modalities, we compare CymbaDiff with existing state-of-the-art semantic scene completion methods that use monocular or stereo RGB inputs. However, we emphasize that our main contribution lies in 3D scene generation. Table 4 compares our method to 3D scene completion methods on the IoU and mIoU metrics reported in their respective publications. All methods are evaluated for 3D semantic scene completion on the SemanticKITTI validation set. The compared methods either use monocular or stereo (image) inputs. Remarkably, despite relying only on input SK and PSA, our method achieves highly competitive performance, matching or exceeding several leading methods that utilize richer input modalities. This demonstrates that SK and PSA offer

Table 4: Quantitative results on the SemanticKITTI validation set. The best results are indicated in **bold**. Mono and Stereo refer to methods using monocular and stereo inputs, respectively, while SK and PSA denote sketch and pseudo-labeled satellite annotations. Note that we demonstrate strong performance using SK+PSA, which are much easier to obtain than images.

| Method | Input | IoU | mIoU | road | sidewalk | parking | other-grnd. | building | car | truck | bicycle | motorcycle | other-veh. | vegetation | trunk | terrain | person | bicyclist | motorcyclist | fence | pole | traf.-sign |
|---|---|---|---|---|---|---|---|---|---|---|---|---|---|---|---|---|---|---|---|---|---|---|
| MonoScene [85] | Mono | 36.9 | 11.1 | 56.5 | 26.7 | 14.3 | 0.5 | 14.1 | 23.03 | 7.0 | 0.6 | 0.5 | 1.5 | 17.9 | 2.8 | 29.6 | 1.9 | 1.2 | 0.0 | 5.8 | 4.1 | 2.3 |
| TPVFormer [86] | Mono | 35.6 | 11.3 | 56.5 | 25.9 | 20.6 | 0.9 | 13.9 | 23.8 | 8.1 | 0.4 | 0.1 | 4.4 | 16.9 | 2.3 | 30.4 | 0.5 | 0.9 | 0.0 | 5.9 | 3.1 | 1.5 |
| NDC-Scene [87] | Mono | 37.2 | 12.7 | 59.2 | 28.2 | **21.4** | 1.7 | 14.9 | 26.3 | 14.8 | **1.7** | **2.4** | 7.7 | 19.1 | 3.5 | 31.0 | 3.6 | 2.7 | 0.0 | 6.7 | 4.5 | 2.7 |
| OccFormer [88] | Mono | 36.5 | 13.5 | 58.9 | 26.9 | 19.6 | 0.3 | 14.4 | 25.1 | **25.5** | 0.8 | 1.2 | 8.5 | 19.6 | 3.9 | 32.6 | 2.8 | 2.8 | 0.0 | 5.6 | 4.3 | 2.9 |
| SparseOcc [89] | Mono | 36.5 | 13.1 | **59.6** | 29.7 | 20.4 | 0.5 | 15.4 | 24.0 | 18.1 | 0.8 | 0.9 | **8.9** | 18.9 | 3.5 | 31.1 | 3.7 | 0.6 | 0.0 | 6.7 | 3.9 | 2.6 |
| IAMSSC [60] | Mono | 44.3 | 12.5 | 54.6 | 25.9 | 16.0 | 0.7 | 17.4 | 26.3 | 8.7 | 0.6 | 0.2 | 5.1 | 24.6 | 5.0 | 30.1 | 1.3 | 3.5 | 0.0 | 6.9 | 6.4 | 3.6 |
| VoxFormer [90] | Stereo | 44.2 | 13.4 | 53.6 | 26.5 | 19.7 | 0.4 | 19.5 | 26.5 | 7.3 | 1.3 | 0.6 | 7.8 | 26.1 | 6.1 | 33.1 | 1.9 | 2.0 | 0.0 | 7.3 | 9.2 | 4.9 |
| DepthSSC [91] | Stereo | **45.8** | 13.3 | 55.4 | 27.0 | 18.8 | 0.9 | 19.2 | 25.9 | 6.0 | 0.4 | 1.2 | 7.5 | 26.4 | 4.5 | 30.2 | 2.6 | **6.3** | 0.0 | 8.5 | 7.4 | 4.1 |
| HASSC-S [92] | Stereo | 44.8 | 13.5 | 57.1 | 28.3 | 15.9 | 1.1 | 19.1 | 27.2 | 9.9 | 0.9 | 0.9 | 5.6 | 25.5 | 6.2 | 32.9 | 2.8 | 4.7 | 0.0 | 6.6 | 7.7 | 4.1 |
| H2GFormer-S [62] | Stereo | 44.6 | 13.7 | 56.1 | 29.1 | 17.8 | 0.5 | 19.7 | 28.2 | 10.0 | 0.5 | 0.5 | 7.4 | 26.3 | 6.8 | 34.4 | 1.5 | 2.9 | 0.0 | 7.2 | 7.9 | 4.7 |
| CymbaDiff | SK+PSA | 43.2 | **14.6** | 52.4 | **33.3** | 13.1 | **10.9** | **32.4** | 32.1 | 0.8 | 1.0 | 0.0 | 3.2 | **28.0** | **8.7** | 22.2 | **4.6** | 4.9 | 0.0 | **11.2** | **12.7** | **5.2** |

a flexible alternative, especially when RGB data are unavailable or impractical, such as in remote sensing.

For semantic scene completion, our method achieves 43.2% IoU and 14.6% mIoU on the SemanticKITTI validation set, outperforming the leading monocular baseline by 1.1% mIoU and the best stereo-based method by 0.9%. This performance gain underscores the strong representational and generative capabilities of CymbaDiff, particularly in reconstructing large-scale structures such as sidewalks, buildings, vegetation, other-ground, and fences. In addition, our method maintains competitive accuracy for smaller objects like people, poles, traffic signs, and tree trunks, demonstrating robustness across a wide range of object sizes and semantic categories. These results collectively highlight the effectiveness and versatility of our approach in diverse urban scene contexts. We present further qualitative examples, including results on underrepresented classes in the Appendix.

# 6 Conclusion

We introduced a novel and scalable task: 3D outdoor semantic scene generation from sketches and pseudo-labeled satellite image annotations. This task offers a low-cost and flexible alternative to traditional annotation-intensive methods, particularly beneficial for applications such as autonomous driving, urban planning. To achieve this, we proposed SketchSem3D, the first publicly available dataset specifically designed for multi-conditioned scene generation in outdoor environments. We proposed CymbaDiff, a diffusion-based generative model designed to enforce structured spatial coherence by explicitly modeling angular continuity and vertical hierarchies, while preserving physical local and global spatial relationships within 3D scenes. CymbaDiff achieves top-tier performance for 3D scene generation and completion using only sparse and abstract input modalities, establishing a solid baseline for future advancements in this field. We hope our new task, dataset, and approach (including code) would foster advancements in related areas.

**Broader Impacts.** CymbaDiff model inherently neutral and designed for positive human-centric applications such as urban simulation and autonomous driving, may pose potential societal risks if misused, particularly in scenarios involving unauthorized mass surveillance.

**Limitations.** While CymbaDiff generates high-quality 3D semantic scenes from freehand (like) sketches and pseudo-labeled satellite image annotations (PSA), obtaining authentic human-drawn sketches could further improve its generalizability and effectiveness in practical human–AI interaction tasks. Future work could focus on using authentic human-drawn sketches for 3D semantic scene generation.

# 7 Acknowledgments

This research was supported by the Australian Government through the Australian Research Council's Discovery Projects funding scheme (project # DP240101926). Professor Ajmal Mian is the recipient of an ARC Future Fellowship Award (project # FT210100268) funded by the Australian Government.

Dr. Naveed Akhtar is a recipient of the ARC Discovery Early Career Researcher Award (project # DE230101058), funded by the Australian Government.

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

# A. Appendix

## A.1 Training Objective

Due to the complexity of the task, our training objective combines multiple loss terms. For the 3D VAE, the objective is:

$$\mathcal{L} = \mathcal{L}_{CE} + \gamma \mathcal{L}_{Lovasz} - \beta D_{KL} \left( q_\phi \left( z | x \right) \| p \left( z \right) \right), \tag{7}$$

where $\gamma = 1.0$ and $\beta = 0.001$ balance the contributions of each loss component, $\mathcal{L}_{CE}$ and $\mathcal{L}_{Lovasz}$ denote the standard cross-entropy and Lovasz-Softmax losses, respectively, following SCPNet [64]. $D_{KL}$ denotes the Kullback–Leibler Divergence between the approximate posterior $q_\phi \left( z | x \right)$ and the prior $p \left( z \right)$, similar to the Latent Diffusion Model (LDM) [1]. The objective function for the CymbaDiff denoising network follows the LDM [1], minimizing the expected squared error between the predicted noise and true noise:

$$\mathcal{L}_{\text{LDM}} = \mathbb{E}_{x, \epsilon \sim N(0,1), t} \left[ \| \epsilon - \epsilon_\theta \left( x_t, t \right) \|_2^2 \right], \tag{8}$$

where $\epsilon_\theta(x_t, t)$ denotes a uniformly-weighted denoising autoencoder applied across time steps $t = 1, \ldots, T$. At each step $t$, the model predicts a denoised estimate of the input $x_t$, which is a noise-corrupted version of the original input $x$.

## A.2 Evaluation Metrics

To evaluate the quality and diversity of the generated 3D semantic scenes, we use two widely used metrics: Fréchet Inception Distance (FID)[74] and Maximum Mean Discrepancy (MMD)[9]. Together, these metrics capture both the statistical similarity and feature-level fidelity between the generated and real data, providing a comprehensive assessment of generative performance. Specifically, FID measures the similarity between the distributions of generated and real samples in a latent feature space. Formally, FID is defined as:

$$\text{FID} = \| M_t - M_g \|_2^2 + Tr \left( C_t + C_g - 2(C_t C_g)^{\frac{1}{2}} \right), \tag{9}$$

where $(M_t, M_g)$ and $(C_t, C_g)$ are the mean and covariance of the real and generated feature distributions. MMD is a non-parametric, kernel-based metric that quantifies the distance between two probability distributions. Unlike FID, MMD does not rely on the assumption that features follow a Gaussian distribution, making it suitable for evaluating generative models under more flexible conditions. In our case, MMD is computed using a Gaussian kernel applied to features extracted from the same latent space as used for FID. The formal definition of MMD is:

$$\text{MMD}^2 \left( X, Y \right) = \mathbb{E}_{x, x'} \left[ k \left( x, x' \right) \right] + \mathbb{E}_{y, y'} \left[ k \left( y, y' \right) \right] - 2 \mathbb{E}_{x, y} \left[ k \left( x, y \right) \right] \tag{10}$$

where $X = \{ x_1, x_2, ..., x_m \}$ and $Y = \{ y_1, y_2, ..., y_m \}$ denote the sets of latent features extracted from real and generated 3D scenes, respectively.

## A.3 Additional Implementation Details

All experiments were conducted on a single NVIDIA GeForce RTX 4090 GPU with 24 GB of RAM. The Variational Autoencoder (VAE) was trained for 22 epochs using the AdamW optimizer with an initial learning rate of 3e-4. The VAE and the CymbaDiff denoising network were trained with a batch size of 2 and 4, each occupying approximately 20 GB of GPU memory. The CymbaDiff denoiser was trained for 31 epochs using the AdamW optimizer with a learning rate of 1e-3 and a weight decay of 1e-4. The number of denoising steps in CymbaDiff was set to 100. A WarmupCosineLR scheduler was used in all training stages to gradually decrease the learning rate, which helped ensure stable convergence.

## A.4 VAE Results

Our CymbaDiff denosing network operates in the latent space of a VAE. To ensure high-quality semantic scene generation, this VAE needs to be accurate. We report the performance of the proposed VAE on the SemanticKITTI validation set in Table 5.

Table 5: VAE reconstruction performance on SemanticKITTI validation set. IoU and mIoU denote Intersection over Union and mean Intersection over Union, respectively.

| Model | Original Spatial Size | Latent Spatial Size | Latent Channel | training epoch | batch size | IoU | mIoU |
|-------|----------------------|---------------------|----------------|----------------|------------|------|------|
| VAE | $256 \times 256 \times 32$ | $64 \times 64 \times 8$ | 8 | 22 | 2 | 92.1 | 92.0 |

Figure 5: Qualitative results on the SemanticKITTI validation set. Columns from the left represent ground truth, and outputs of CymbaDiff (our method), MonoScene, OccFormer, and VoxFormer.

## A.5 Effieicency Comparison

we provide quantitative comparisons in the Table 6 across methods in terms of parameter count and runtime performance. These results demonstrate that CymbaDiff achieves a favorable trade-off between model efficiency and computational cost, offering competitive performance with significantly fewer parameters compared to these two generative models.

## A.6 Cross-domain Test

We have now trained SemCity[11] and CityDreamer[93]on the SketchSem3D dataset to compare with our CymbaDiff. To ensure compatibility with our 3D voxel-based setup, we integrated their denoisers into our framework. We also attempted to train the full SemCity pipeline directly, but it resulted in unstable training, with the VAE loss diverging to NaN, an issue also reported by other users on SemCity's official GitHub page. Please note, CityDreamer is designed for 2D generation and cannot be directly applied to 3D voxel scenes. As shown in the Table 7, CymbaDiff consistently outperforms both baselines across all evaluation metrics strongly.

The reason why Semcity and CityDreamer do not perform well in our experiments is their denoisers (provided in their official GitHub repositories). The denoiser in SemCity only has convolutional and linear layers, whereas that in CityDreamer relies on a simple stacking of transformer layers. Although transformer layers can model long-range dependencies, such simplified designs may be suboptimal for large-scale 3D voxel scene generation, where sparse and irregular data demand specialized mechanisms to effectively capture both local geometry and relevant global context.

## A.7 Qualitative results on 3D Semantic Scene Completion

To demonstrate the effectiveness of our proposed framework for 3D semantic scene completion, we present additional qualitative results in Figures 5. The figure displays representative examples randomly selected from the SemanticKITTI validation set [68]. CymbaDiff accurately delineates fine-grained boundaries of 3D scenes and objects by incorporating the cylinder Mamba blocks, which

Table 6: Efficiency comparison. M: Million, and S: seconds.

| Input Modality | Parameters (M) | Inference Times (S) |
|---|---|---|
| 3D DIT | 195 | 4.5 |
| 3D Latent Diffusion | 1265 | 11.4 |
| CymbaDiff | 23 | 7.2 |

Table 7: Cross-domain Comparison

| Method | Sketch-based SemanticKITTI FID ↓ | Sketch-based SemanticKITTI MMD ↓ | Sketch-based KITTI-360 FID ↓ | Sketch-based KITTI-360 MMD ↓ |
|---|---|---|---|---|
| 3D SemCity[11] | 987.91 | 0.26 | 740.09 | 0.25 |
| 3D CityDreamer[93] | 950.16 | 0.26 | 754.47 | 0.25 |
| CymbaDiff | 40.67 | 0.04 | 107.53 | 0.08 |

promotes structured spatial coherence through explicit modeling of angular continuity and vertical hierarchies.

### A.8 Licenses

**Licenses of SemanticKITTI and SSCBench KITTI-360.** The SemanticKITTI dataset is licensed under the CC BY-NC-SA 4.0, while the SSCBench KITTI-360 dataset is released under CC BY-NC-SA 3.0 license.

**Terms of Use and License of SketchSem3D.** The SketchSem3D dataset is licensed under CC BY-NC-SA 4.0.

