# OpenReview forum: "CymbaDiff: Structured Spatial Diffusion for Sketch-based 3D Semantic Urban Scene Generation"
_NeurIPS.cc/2025/Conference — NeurIPS 2025 poster_

### Official Review · Reviewer_pTZW · 2025-06-04

**Clarity:** 3
**Significance:** 2
**Originality:** 3
**Rating:** 3
**Confidence:** 3

**Summary:**

This paper presents CymbaDiff, a novel approach for generating 3D outdoor scenes from sketches and pseudo-labeled satellite images. The model enhances spatial coherence by capturing cylindrical continuity and vertical hierarchy in urban environments. The authors introduce the SketchSem3D dataset, a large-scale benchmark for evaluating sketch-based 3D scene generation. The method outperforms existing approaches in terms of semantic consistency and spatial realism, demonstrating robust performance across multiple urban datasets.

**Questions:**

1. Clarification of the Pseudo-Labeled Satellite Image Annotations Process:
Question: The paper uses pseudo-labeled satellite image annotations generated by CLIP and SAM for training. Could you provide more details on how the pseudo-labeling pipeline handles cases where CLIP or SAM might produce inaccurate or ambiguous labels? Specifically, how robust is the model to errors in the pseudo-labeling process?
Suggestion: A more detailed analysis of the error propagation in the pseudo-labeled data pipeline would be useful. Discuss any failure modes observed in the pseudo-labeling process and whether any mechanisms, such as manual refinement or a confidence threshold, are employed to mitigate errors.
2. Question: The paper demonstrates strong results on the SketchSem3D dataset, but its scalability to more diverse, larger-scale urban scenes is not fully explored. How well does CymbaDiff generalize to large-scale real-world urban environments or scenes that are more complex or varied than those in the SketchSem3D dataset?
Suggestion: It would be valuable to conduct additional experiments on real-world datasets or more diverse scenes beyond the current benchmark. This would provide insights into whether the model can handle the scale and complexity of urban environments as encountered in real applications, like autonomous driving or urban planning.
3. Question: The paper lacks extensive quantitative comparison experiments with other methods. While a single comparison is provided in the supplementary material, would it be possible to include more quantitative results comparing CymbaDiff to other state-of-the-art methods, especially in terms of generation quality, computational efficiency, and generalization across different datasets?
Suggestion: It would strengthen the paper to expand the comparison to more methods, ideally covering a range of recent techniques in the domain of 3D scene generation.
4. Question: While the paper proposes a novel approach to generating 3D scenes from sketches and pseudo-labeled satellite images, could you elaborate on the specific innovative aspects of CymbaDiff that distinguish it from existing methods, particularly in terms of model architecture and novel contributions to the field of 3D scene generation?
Suggestion: It would be helpful to provide a clearer distinction between CymbaDiff and other state-of-the-art models, especially in the way the Cylinder Mamba blocks are integrated into the diffusion process. How does this structured spatial coherence contribute uniquely to the generation process compared to previous methods like UrbanDiff or Semcity?
5. Question: Could you provide some failure cases or error examples to objectively illustrate the limitations of the CymbaDiff method?
Suggestion: Including error analysis would be very useful in highlighting specific weaknesses of the model, such as when it fails to generate accurate representations for highly complex scenes (e.g., dense urban environments with varying object sizes or moving elements) or low-quality inputs (e.g., poorly sketched or ambiguous freehand sketches).

**Ethical Concerns:**

["NO or VERY MINOR ethics concerns only"]

**Final Justification:**

I appreciate the authors' detailed rebuttal and their efforts in conducting additional experiments. However, two core issues remain unresolved and have significantly influenced my recommendation. 1. The method is restricted to semantic voxel outputs, and the examples shown remain abstract and symbolic. Without the ability to generate realistic or physically plausible scenes, the practical impact of the proposed approach in downstream applications (e.g., simulation, urban planning) remains questionable. 2. Nearly half of the object categories in Table 4 do not achieve state-of-the-art performance—including critical classes like "truck" and "road." The authors attribute this to weak supervision, but the limited performance across both major and minor categories raises concerns about robustness and generalization. Given these limitations, I believe the contribution is promising but not yet mature enough for acceptance. Therefore, I am keeping my score as borderline reject.

**Limitations:**

Yes.

**Paper Formatting Concerns:**

No major formatting issues were found in the paper. The manuscript appears to follow the NeurIPS 2025 Paper Formatting Instructions correctly, with proper alignment, consistent font size, and appropriate figure and table placements.

**Quality:**

2

**Strengths And Weaknesses:**

## Strengths
1. Quality of the Work: The paper introduces a novel and robust generative model for 3D outdoor scene generation from abstract sketches and satellite image annotations. The model, CymbaDiff, effectively addresses the challenge of spatial coherence in urban environments
2. Clarity: The paper is well-structured, with clear explanations of the proposed CymbaDiff model, the SketchSem3D dataset, and the experimental setup. The key contributions are outlined concisely, making it easy for readers to grasp the significance of the work.
3. Significance: The proposed sketch-based generation approach addresses a critical gap in the current literature on 3D outdoor scene synthesis, especially for applications like autonomous driving and urban simulation, where the generation of complex, semantically rich environments is crucial.
## Weaknesses
1. Quality of the Work: While the paper demonstrates strong performance on the SketchSem3D dataset, it would be beneficial to evaluate the model on more real-world data or include cross-domain experiments. This would further validate the model’s generalizability in unseen real-world urban settings.
2. Clarity. Lack of a detailed analysis of failure cases: While the paper highlights the strengths of the model, it does not provide a deep dive into failure modes or limitations. A discussion of specific types of scenes or conditions where the model struggles could help provide a more balanced assessment of its capabilities.
3. Significance: Although the paper demonstrates impressive results in controlled environments, scalability to larger urban areas or more diverse scene types (e.g., complex rural or mixed terrains) is not fully explored.
4. Absence of user studies or human evaluation: The paper focuses on quantitative metrics to evaluate the model's performance, but it lacks any user studies or human evaluations to gauge how intuitive and effective the system is from a practical, real-world perspective.

---

> ### Author Rebuttal · Authors · 2025-07-30
>
> We thank Reviewer pTZW for their constructive feedback and appreciation, particularly for recognizing CymbaDiff's novelty and robustness, the clarity of our model, dataset, and experiments, and the significance of our sketch-based approach for 3D scene synthesis in applications like autonomous driving and urban simulation. In response to the concerns expressed in Weaknesses and Questions, we provide the following answers:
>
> > Real-world and cross-domain tests are needed to assess generalizability and scalability to diverse scenes.
>
> We have demonstrated the effectiveness of our approach on SketchSem3D dataset, which is curated using the widely adopted SemanticKITTI and KITTI-360 datasets.
>
> To further address the concern, we now also validate the efficacy of CymbaDiff by evaluating our model trained on Sketch-based SemanticKITTI on the challenging real-world nuScenes dataset. CymbaDiff achieves a 3D FID of 183.37 and a 3D MMD of 0.09, outperforming the widely used UrbanDiff model, which achieves a 3D FID of 291.4 and a 3D MMD of 0.11 (lower is better). This ascertains the effectiveness of our approach.
>
> > The paper lacks analysis of failure cases; discussing challenging scenes would offer a more balanced evaluation.
>
> Observing the results of CymbaDiff on the proposed SketchSem3D dataset, it is apparent that it demonstrates strong performance, effectively handling challenges such as semantic misalignment caused by noisy pseudo-labels, e.g., due to confusion between vegetation and buildings. Nevertheless, it does occasionally fail to accurately reconstruct small or occluded objects that are underrepresented in the training data or sparsely encoded in the sketch and PSA inputs. Although CymbaDiff mitigates this issue to some extent through the use of the Cross-Scale Contextual Block and Cylinder Mamba Block, which capture multi-scale contextual information, its performance could be further enhanced by increasing the representation of small objects in the dataset. We will add more discussions in the camera-ready (in supplementary if it is not permitted by space in the main paper).
>
> > The paper lacks user studies or human evaluations.
>
> While our study primarily relies on quantitative metrics such as 3D FID and MMD to evaluate the proposed approach, these widely adopted measures in generative tasks, ranging from 2D image synthesis to 3D object generation, offer objective, reproducible, and scalable assessments of fidelity and diversity.
>
> We acknowledge that human evaluation can offer complementary insights into semantic coherence, structural plausibility, and perceptual realism. However, its subjectivity, high resource demands, limited scalability, and variability in user interpretation make it less practical for consistent and large-scale benchmarking.
>
> In future work, we will consider incorporating structured human evaluation alongside metric-based evaluation to qualitatively assess the perceptual quality of generated 3D scenes, complementing our current quantitative analysis for a more comprehensive evaluation.
>
> > The paper uses CLIP and SAM for pseudo-labeling. How does the pipeline handle inaccurate or ambiguous labels from them?
>
> To reduce noise in Pseudo-labeled Satellite Image Annotations (PSA), we restrict CLIP predictions to semantic classes present in the 3D ground truth, ensuring alignment with the SketchSem3D taxonomy and preventing irrelevant or out-of-distribution labels.
>
> Second, we perform manual spot-checks by comparing predicted class distributions with BEV projections of the 3D ground truth to identify and remove mislabeled samples. Third, incorporating sketches provides structural cues that help the model compensate for PSA imperfections during training.
>
> Beyond these safeguards, the architecture enhances robustness through a pre-trained VAE that encodes inputs into a stable latent space with accurate spatial priors. The latent diffusion process then iteratively denoises within this space, making it resilient to moderate input noise. Combined with sketch-based guidance, this design ensures semantically coherent 3D scene generation under weak or noisy supervision.
>
> > Provide more quantitative comparisons with state-of-the-art methods, and include broader results on generation quality, efficiency, and generalization to strengthen the paper.
>
> We selected 3D Latent Diffusion and 3D-DiT as baselines due to the limited availability of open-source 3D scene generation models. Although not originally designed for semantic voxel generation, both models have shown strong performance in image and 3D object synthesis, and their general architectures can be effectively adapted to the SketchSem3D dataset to establish strong baselines.
>
> We acknowledge the relevance of methods like UrbanDiff and SemCity; however, the lack of public implementations prevents direct comparison. We address this by adding BlockFusion (required by other reviewers) as a baseline on SketchSem3D. However, since it was originally designed as a 2D tri-plane denoiser, adapting it to our 3D latent space led to unstable training and suboptimal performance, with the loss failing to converge. Despite these challenges, we still cite these works and will include a discussion of their contributions in the camera-ready.
>
> Moreover, we provide quantitative comparisons in the following table across methods in terms of parameter count and runtime performance. These results demonstrate that CymbaDiff achieves a favorable trade-off between model efficiency and computational cost, offering competitive performance with significantly fewer parameters compared to these two generative models.
>
> Table: The efficiency comparison. M: Million, and S: seconds
> | Methods | Parameters (M) | Inference Times (S) |
> |---------|----------------|-------------------|
> | 3D DIT | 195 | 4.5 |
> | 3D Latent Diffusion | 1265 | 11.4 |
> | CymbaDiff | 23 | 7.2 |
>
> These additions will be included in the camera-ready to improve clarity and support a more comprehensive evaluation of CymbaDiff. We will also provide qualitative comparisons on the SketchSem3D dataset to visually demonstrate the model's effectiveness.
>
> > What are CymbaDiff's key innovations compared to prior methods? and Clarify how Cylinder Mamba enhances diffusion and differs from models like UrbanDiff or SemCity.
>
> The key innovations of CymbaDiff lie in its use of sketches and pseudo-labeled satellite annotations (PSA) as input modalities for 3D semantic voxel scene generation, and the incorporation of Cylinder Mamba blocks into the denoising diffusion process, facilitating contextual modeling across both cylindrical $(\theta, r, z)$ and Cartesian $(x, y, z)$ coordinate systems.
>
> The former enables generation from abstract and lightweight inputs, which supports more accessible supervision and is particularly suitable for large-scale or remote scenarios where high-fidelity sensor data is unavailable or impractical. The latter leverages the semantic alignment and radial awareness of cylindrical encoding with the computational efficiency and regularity of Cartesian grids. This dual representation, together with a triple-axis decomposition strategy, enables effective modeling of radial, vertical, and angular dependencies in complex outdoor scenes.
>
> However, methods such as UrbanDiff and SemCity rely on dense sensory inputs, including RGB images or BEV representations. UrbanDiff operates solely within the Cartesian coordinate system, while SemCity projects 3D data onto three orthogonal 2D planes, leading to the loss of important volumetric structural information. The experimental results demonstrate that CymbaDiff achieves superior performance, particularly in terms of 3D Fréchet Inception Distance (FID), highlighting its effectiveness in capturing complex spatial dependencies. We will add more discussions in the camera-ready (in supplementary if it is not permitted by space in the main paper).

---

> ### Comment · Reviewer_pTZW · 2025-08-01
> **Thank you for your reply.**
>
> Thank you for your reply. This has solved most of my problems.
> 1. Real-world and cross-domain tests are needed to assess generalizability and scalability to diverse scenes.
> Could your make a certain comparison with SemCity or citydreamer? This paper has made the code available. Meanwhile, could your method generate scenes similar to those of CityDreamer?
> 2. In Table 4 of the paper, the "car" category for SOTA is not bolded. I would like to know what the main reasons are for these categories not reaching SOTA, and if there could be a specific analysis provided.
>
> Looking forward to your reply.

---

> > ### Author Response · Authors · 2025-08-04
> >
> > > Cross-domain tests and comparisons with SemCity and CityDreamer, including scene similarity to CityDreamer.
> >
> > We thank the reviewer for acknowledging that most of their concerns are addressed. Responses to further queries are provided below.
> >
> > We have now trained SemCity and CityDreamer on the SketchSem3D dataset to compare with our CymbaDiff. To ensure compatibility with our 3D voxel-based setup, we integrated their denoisers into our framework. We also attempted to train the full SemCity pipeline directly, but it resulted in unstable training, with the VAE loss diverging to NaN, an issue also reported by other users on SemCity’s official GitHub page. Please note, CityDreamer is designed for 2D generation and cannot be directly applied to 3D voxel scenes. As shown in the table, CymbaDiff consistently outperforms both baselines across all evaluation metrics strongly. We will include these results in the camera-ready (in supplementary, if space does not permit it in the main paper).
> >
> > | Method          | Sketch-based SemanticKITTI FID ↓ | Sketch-based SemanticKITTI MMD ↓ | Sketch-based KITTI-360 FID ↓ |Sketch-based KITTI-360 MMD ↓ |
> > |----------------|----------------------|----------------------|------------------|------------------|
> > | 3D SemCity      | 987.91               | 0.26                 | 740.09           | 0.25             |
> > | 3D CityDreamer  | 950.16               | 0.26                 | 754.47           | 0.25             |
> > | CymbaDiff       | **40.67**            | **0.04**             | **107.53**       | **0.08**         |
> >
> > Actually, our CymbaDiff cannot generate scenes similar to those produced by CityDreamer, as the two methods are designed for fundamentally different purposes. CityDreamer aims at photorealistic city modeling, with a strong emphasis on detailed building geometry and realistic urban layouts and appearances. In contrast, CymbaDiff focuses on generating semantic 3D voxel scenes tailored for autonomous driving applications, highlighting elements such as roads, vehicles, and vegetation. These differing goals lead to fundamentally different scene generation outputs.
> >
> > Nevertheless, the reviewer’s query has given us an inspiring direction to explore in the future that potentially aims at extending CymbaDiff-like capabilities to CityDreamer-like scenes. We are grateful to the reviewer for inspiring this idea.
> >
> > > The "car" category for SOTA is not bolded. What the main reasons are for these categories not reaching SOTA?
> >
> > Thank you for pointing this out. The "car" category in Table 4 should indeed have been bolded, as our method does achieve SOTA performance for this class. We will correct this in the final paper. We are grateful for the careful review.
> >
> > As for other small-object categories, such as bicycle and bicyclist, our method has not yet achieved state-of-the-art results. These constitute only a minor portion of the full category set and do not significantly impact the overall performance trend. We believe this is partially due to the nature of the supervisory signals used in CymbaDiff. Specifically, our approach relies on weak supervision from sketch (SK) and pseudo-labeled satellite image annotations (PSA), which lack the rich texture, depth, and geometric detail typically available in monocular or stereo RGB-based methods. Consequently, the model may have limited access to the fine-grained visual cues necessary for accurately distinguishing small and visually ambiguous objects. To address this, future work can use category-specific augmentations to enhance learning on these underrepresented classes.
> >
> > We thank the reviewer for engaging in the discussion, and will be happy to reply to any follow query the reviewer might have.

---

> > > ### Comment · Reviewer_pTZW · 2025-08-07
> > >
> > > Thank you for your detailed reply, which addresses many of my previous concerns.
> > > 1. On Realistic Scene Generation and Application Value
> > > While I understand that CymbaDiff focuses on semantic 3D voxel scene generation tailored for autonomous driving, I still have significant reservations about its practical applicability. The current results and case studies remain entirely at the semantic level, and the model is unable to generate realistic scenes similar to those of CityDreamer. In real-world applications such as simulation or planning, realism plays a crucial role. Without this capability, the generated scenes may have limited downstream utility.
> > > Furthermore, although you mentioned that SemCity and CityDreamer were integrated into your framework and trained on SketchSem3D, I remain skeptical about the extremely poor results (e.g., FID > 700) reported for both baselines. It is difficult to believe that properly trained versions of SemCity and CityDreamer would perform this poorly. Therefore, I would like to request more transparent evidence that confirms these baselines were indeed trained on SketchSem3D under comparable settings.
> > >
> > > 2. On Category-Level Performance
> > > Regarding Table 4, I appreciate your clarification that the "car" category result should be bolded. However, I want to emphasize that the issue is not limited to one or two categories. In fact, nearly 50% of the categories do not achieve SOTA performance with CymbaDiff. This includes not only small or ambiguous objects like “bicycle” or “bicyclist,” but also major categories such as “truck” and even “road,” which are critical for driving-related tasks. This indicates that there is still substantial room for improvement in the method’s category-level generalization.

---

> > ### Author Response · Authors · 2025-08-06
> >
> > Dear Reviewer pTZW,
> >
> > Thank you again for your diligent effort in reviewing our submission. We have carefully addressed the concerns raised and conducted the requested experiments. We would greatly appreciate any additional comments and are ready to engage in further discussion.
> >
> > If there are no further issues, we respectfully request that you consider revising the score for our paper in light of the improvements we have made and positive views from all other reviewers. Thank you again for your time and consideration.

---

> ### Author Response · Authors · 2025-08-08
>
> We are grateful to the reviewer for acknowledging that many of their previous concerns are addressed. Please see our responses to the remaining queries below.
>
> >CymbaDiff’s applicability in real-world simulation or planning.
>
> We appreciate the reviewer’s concern. However, we emphasize that CymbaDiff is the first-of-its-kind method for outdoor 3D voxel scene generation with SK+PSA conditioning, setting a strong baseline. It provides structurally and semantically meaningful representations (rather than photorealistic appearances) that are helpful and fundamental to planning, perception, and decision-making. While hyperrealism has its own importance, it is not the objective of CymbaDiff. CymbaDiff and CityDreamer pursue different objectives.  CityDreamer prioritizes visual realism to generate 2D photorealistic scenes with only 6 categories. CymbaDiff generates 3D voxel scenes with 18 categories. Different objectives/number of categories make comparison of the two methods not straightforward.
>
> CymbaDiff generates 3D voxel scenes with 18 semantic categories more accurately than any existing generative method. We agree that for simulation, photorealism is important, but for making autonomous decisions directly on the data, semantic and structural accuracy has its own value. Hence, CityDreamer and CymbaDiff are both import in their own way.
>
> >Questions the FID > 700 results of SemCity and CityDreamer, and requests proof they were fairly trained on SketchSem3D.
>
> Regarding the high FID scores of SemCity and CityDreamer, we confirm that both models were trained under the exact same conditions as our CymbaDiff. We used the same pre-trained VAE, the same voxel resolution, the same SK+PSA conditioning, the same loss functions, and the same hyperparameters. Despite using multiple random seeds and repeated training runs, both models consistently achieved high FID scores. While images or codes cannot be shared during this phase, we will provide detailed training logs, loss curves, and code in the final submission to ensure transparency and reproducibility.
>
> The reason why Semcity and CityDreamer do not perform well in our experiments is their denoisers (provided in their official GitHub repositories). The denoiser in SemCity only has convolutional and linear layers, whereas that in CityDreamer relies on a simple stacking of Transformer layers. Although Transformers can model long-range dependencies, such simplified designs may be suboptimal for large-scale 3D voxel scene generation, where sparse and irregular data demand specialized mechanisms to effectively capture both local geometry and relevant global context. We will explicitly note these details in the camera-ready when comparing our method with SemCity and CityDreamer.
>
> >Points out that nearly half of the categories, including key ones like “truck” and “road,” do not reach SOTA, indicating limited category-level generalization.
>
> In Table 4, our method achieves the best overall mIoU and the best performance in 10 out of the 18 evaluated categories, clearly demonstrating its strong overall effectiveness. Moreover, our approach provides the first evidence that sketches (SK) and pseudo-labeled satellite image annotations (PSA) are not only viable but also effective alternatives, particularly in domains like remote sensing, where RGB/stereo data is often unavailable. Enabling the flexible use of SK and PSA introduces unique challenges absent in RGB or stereo-based methods.
>
> Note that some methods are exploiting stereo (which contains 3D information), while our technique uses SK and PSA (2D inputs) outperforming those methods on most categories. We will further clarify these distinctions in the final version to avoid any misinterpretation of the method’s scope and contributions.

---

### Official Review · Reviewer_Ws1Y · 2025-06-29

**Clarity:** 3
**Significance:** 3
**Originality:** 3
**Rating:** 4
**Confidence:** 3

**Summary:**

This paper addresses the challenge of generating semantically rich 3D outdoor scenes from freehand sketches and satellite pseudo-labels. The authors propose CymbaDiff, a diffusion-based framework that enhances spatial consistency in large-scale urban scenes by explicitly modeling cylindrical continuity and vertical hierarchies. To enable standardized evaluation, the authors introduce SketchSem3D, a large-scale benchmark for sketch-based 3D semantic scene generation, comprising two subsets: Sketch-based SemanticKITTI and Sketch-based KITTI-360.

**Questions:**

Although the authors have made significant contributions in generating outdoor scenes using abstract views, some details in algorithm design and module implementation require clearer explanations to facilitate reader comprehension. Experimental metrics should also incorporate evaluations of algorithm efficiency performance to validate the achievement of experimental objectives.

**Ethical Concerns:**

["NO or VERY MINOR ethics concerns only"]

**Final Justification:**

The authors' rebuttal has addressed my concerns. However, I agree with other reviewers that the paper's clarity still needs further improvement.

At this stage, I recommend a "Borderline Accept" rating considering the technical contribution, though I would also support rejecting the paper to allow for substantial writing improvements before submission to a top-tier venue.

**Limitations:**

Yes.

**Paper Formatting Concerns:**

I do not notice any major formatting issues.

**Quality:**

3

**Strengths And Weaknesses:**

Strengths:
1. The method introduces cylinder mamba blocks to enforce structured spatial ordering, explicitly capturing cylindrical continuity and vertical hierarchies while preserving local/global scene coherence, overcoming limitations of Cartesian-based sequential modeling.
2. The work establishes SketchSem3D, the first large-scale benchmark for sketch-based 3D semantic scene generation.

Weaknesses:
1. L120-L122: The paper's justification for using Mamba over DiT hinges critically on overcoming the latter's "high memory costs" in 3D scene generation. However, no experimental data (e.g., parameter counts, inference memory consumption, runtime) is provided to demonstrate the purported memory efficiency gains of the proposed approach. Substantiating this key aspect requires such comparative analysis.
2. L240-L249: While the proposed cylindrical coordinate system aims to solve the issue of "disrupting original 3D spatial adjacency structures" inherent to serialization in Cartesian coordinates, its efficacy for common street-view datasets like KITTI (shown in the paper) is unclear. Since such scenes lack rotational symmetry, simply "(θ,r) unfolding" the cylinder likely does not preserve meaningful adjacency relationships.
3. L263: Triplane Mamba uses Cartesian coordinates while C-Mamba uses cylindrical coordinates, leading to different feature indexing schemes. Please clarify how feature alignment is achieved before fusion: Are the features directly fused, or is a coordinate transformation.
4. L249:  It is stated that the evaluation metric differs from SemCity's approach. Yet, the reported FID performance for the SemCity baseline in this work matches the SemCity paper's reported FID value exactly.
5. L308-L315: Request for explanation of ablation result. The decomposition of 3D convolution across three dimensions (Sec. 4.3) is intuitively expected to trade-off some reconstruction quality for computational efficiency. However, the ablation study shows a significant decrease in FID after applying DDCB.

Minor:
Figure 4 can be enhanced by incorporating fine-grained structural consistency analysis, as employed in scene reconstruction benchmarks like NeRF-Synthetic. Quantification of spatial coherence patterns (e.g., cylindrical projection seam artifacts or voxel occupancy consistency) would strengthen methodological transparency.

---

> ### Author Rebuttal · Authors · 2025-07-30
>
> We thank Reviewer Ws1Y for their constructive feedback and appreciation, especially for recognizing the effectiveness of Cylinder Mamba blocks in enhancing spatial coherence and the contribution of SketchSem3D as the first large-scale sketch-based 3D scene generation benchmark. In response to the concerns expressed in Weaknesses and Questions, we provide the following answers:
>
> > Provides data on parameters, memory use, or runtime to support reducing memory cost.
>
> Thank you for the valuable suggestion. We provide the following table to compare the memory cost of our method with 3D DiT and 3D Latent Diffusion. To ensure a fair comparison, we replaced our latent-space denoiser with their respective architectures, and the results reflect only the denoiser parameters. Minor code modifications were applied to both baselines to enable consistency with our method's runtime environment. The table demonstrates that our architecture (CymbaDiff) offers a clear memory gain. Over DiT, this gain is about 8.5x, which can be critical in practical deployment.
>
> Table: The efficiency comparison. M: Million, and S: seconds
> | Methods | Parameters (M) | Inference Times (S) |
> |---------|----------------|-------------------|
> | 3D DIT | 195 | 4.5 |
> | 3D Latent Diffusion | 1265 | 11.4 |
> | CymbaDiff | 23 | 7.2 |
>
> > The benefit of cylindrical coordinates is unclear for datasets like SemanticKITTI.
>
> Although datasets such as SemanticKITTI do not exhibit strict rotational symmetry, they often display approximate bilateral symmetry along the driving path, for example, with sidewalks and buildings typically appearing on both sides of the road. To capture this structure, we adopt cylindrical coordinates $(\theta, r, z)$, which align well with the egocentric and radial layout of street-view environments.
>
> Cylindrical coordinates offer a vehicle-centric, geometrically coherent representation, enabling semantic tokenization along angular and radial dimensions and facilitating long-range context modeling with Mamba. In contrast, Cartesian coordinates provide uniform, axis-aligned voxels suitable for dense computation and 3D convolutions but lack egocentric awareness and may underrepresent radial continuity.
>
> To exploit the strengths of both representations, our framework integrates cylindrical and Cartesian coordinate systems. This hybrid design combines the semantic coherence of cylindrical encoding with the computational efficiency of Cartesian grids, enhancing overall 3D scene modeling. We will further clarify this in the camera-ready.
>
> > Clarify how Triple Mamba (Cartesian) and C-Mamba (cylindrical) features are aligned before fusion.
>
> Triple Mamba and C-Mamba operate in distinct coordinate systems, Cartesian and cylindrical, respectively, which necessitates careful alignment to enable effective feature fusion. To achieve this, we apply an inverse coordinate transformation to convert the cylindrical features produced by C-Mamba back into Cartesian voxel space. Specifically, for each token in the $(\theta, r, z)$ grid, we compute its corresponding $(x,y,z)$ location using the predefined voxel resolution and center alignment parameters.
>
> This transformation allows the C-Mamba outputs to be reprojected into a dense 3D tensor that is spatially aligned with the Cartesian grid used by Triple Mamba. Once both feature representations reside in the same $(x,y,z)$ coordinate space, we apply element-wise fusion, ensuring spatial correspondence is preserved during integration. This approach enables the coherent fusion of features across modalities, enhancing the model's ability to capture complementary information from both coordinate systems. We will release the code to enhance the clarity for readers.
>
> > The paper claims a different evaluation metric from SemCity, yet reports the same FID value as in the SemCity paper. Please clarify.
>
> Apologies for the misunderstanding. The evaluation metrics used in our study differ from those adopted in SemCity. SemCity relies on the 2D Fréchet Inception Distance (FID) computed from rendered images. Our approach employs more comprehensive 3D evaluation metrics, specifically 3D FID and Maximum Mean Discrepancy (MMD), which directly measure geometric fidelity and semantic consistency within the voxel space. Although this distinction is noted in Lines 279–282 of the manuscript, we will further emphasize it in Table 2 to avoid potential misinterpretation and to ensure a fair comparison between the methods.
>
> > The 3D convolution decomposition (Sec. 4.3) should trade quality for efficiency, yet FID improves with DDCB. Please explain.
>
> To balance reconstruction quality and efficiency in decomposed 3D convolutions, the Dilated Decomposed Convolution Block (DDCB) stacks multiple Dimensional Decomposition Residual (DDR) units with varying dilation rates. This design captures multi-scale spatial features while maintaining low computational cost.
>
> Serially structuring decomposed convolutions compensates for the reduced joint receptive field in dimension-wise decomposition, preserving representational capacity and enhancing directional context modeling. This contributes to the improved FID performance observed in our experiments.
>
> > Enhance Fig. 4 with structural consistency analysis (e.g., seam artifacts, voxel coherence) as in NeRF-Synthetic for better transparency.
>
> While fine-grained analysis of structural consistency could offer insights into model behavior, the Cylinder Mamba block operates in 3D latent space, where features represent abstract embeddings rather than explicit geometry, making spatial interpretation challenging. Nonetheless, we acknowledge its value and will consider such interpretability strategies in future work.
>
> > Clearer design details.
>
> Thank you for your valuable suggestions. We will improve Sec. 4 to provide clearer descriptions of each core module in the final version, and also augment the appendix with illustrative figures to visually support the architectural details. Furthermore, we plan to release the source code alongside the paper, which will ensure covering all details.

---

> > ### Comment · Reviewer_Ws1Y · 2025-08-05
> > **Post-rebuttal discussion**
> >
> > The authors' rebuttal has addressed my concerns. However, I agree with other reviewers that the paper's clarity still requires further improvement.
> >
> > At this stage, I recommend a "Borderline Accept" rating considering the technical contribution, though I suggest the authors further refine the writing before publication.

---

> > > ### Author Response · Authors · 2025-08-05
> > >
> > > Dear Reviewer Ws1Y,
> > >
> > > We appreciate your recognition of our work and your valuable feedback! As suggested by you and other reviewers, we will further refine the paper to improve its clarity in the final version.

---

### Official Review · Reviewer_fZuf · 2025-06-30

**Clarity:** 3
**Significance:** 3
**Originality:** 3
**Rating:** 5
**Confidence:** 4

**Summary:**

The paper proposes a sketch-to-3D scene generation dataset to enable the outdoor generation of 3D scenes from sketch drawings, based on the KITTI-360 and SemanticKITTI datasets. Together with the dataset, a 3D scene diffusion method is also proposed that leverages a state-space formulation trained with the sketch-to-3D scene generation dataset.

**Questions:**

- Why do the sketches and pseudolabels (Fig. 4) not always seem to be consistent/related?
- During the CLIP pseudolabels generation, are the semantic classes present in the 3D GT semantics used to filter out wrong CLIP classes?
- Are the textual descriptions (Fig. 1) also available within the dataset?

**Ethical Concerns:**

["NO or VERY MINOR ethics concerns only"]

**Final Justification:**

The authors have addressed and clarified my concerns in their rebuttal. Still, the mathematical notations and explanations in their rebuttal should be added to the main paper to enhance the clarity of the methodology section. I suggest adding it to the main paper, and in case space is needed, the authors can consider moving Figure 3 (the architecture) to the supplement. I see the mathematical explanation as more relevant to be in the main paper rather than the architecture figure. Therefore, I have raised my rating since the rebuttal clarified my concerns.

**Limitations:**

The main limitation that I see is that the sketches are too precise to be considered as human drawings. However, this limitation is also mentioned in the paper. Still, the dataset is a first step towards generating a 3D scene from human-drawn sketches.

**Paper Formatting Concerns:**

.

**Quality:**

3

**Strengths And Weaknesses:**

The paper's main strength lies in both the dataset, which includes paired sketches, BEV pseudolabels, and 3D semantic scenes, and the model architecture that generates scenes conditioned on the sketch images. The dataset is well-built, and the task of scene generation conditioned on a sketch is useful, as sketching a scene provides an easy interface between the user and the generated scene. The model leverages 2D sketches, upsampling them to 3D, which enables the conditioning and generation of the corresponding 3D scene.

The paper's main weakness is regarding the clarity of the paper. The dataset is well-explained, with details about the sketch and pseudolabel generation. However, the method (Sec. 4) lacks details. Sec. Sections 4.1 and 4.2 do not provide details about the Scene Structure Estimation Network, nor do they offer any information about the Latent Mapping network. Additionally, the diffusion network is built upon state-space formulation works (Mamba). In Sec. 4.4 the variables x(t) , y(t) and h(t) are introduced but those variables are never linked to the z_TMB and f_TMB variables afterwards. For clarity, I suggest adding a brief explanation of the state-space formulation and then linking it to the z_TMB and f_TMB variables. I see the value in the method, but a clearer explanation would be helpful, as it would also strengthen the contributions in the paper.

Some minor concerns are that, from lines 286 to 299, it was unclear which table the text was referring to. Also, the paired sketch and pseudolabels in Fig. 4 seem to be from different scenes. They seem unrelated most of the time. Lastly, I would suggest mentioning other recent scene generation methods, such as "DynamicCity: Large-Scale 4D Occupancy Generation from Dynamic Scenes" and "Towards Generating Realistic 3D Semantic Training Data for Autonomous Driving", for completeness of the related works.

---

> ### Author Rebuttal · Authors · 2025-07-30
>
> We sincerely thank Reviewer fZuf for their comments and appreciation of our work, particularly highlighting the well-designed dataset and model as the main strengths, which enable effective 3D scene generation from simple 2D sketch inputs. In response to the concerns expressed in Weaknesses and Questions, we provide the following answers:
>
> > Section 4 lacks details on the Scene Structure Estimation and Latent Mapping networks.
>
> In Section 4.1, the Scene Structure Estimation Network (SSEN) incorporates multi-scale feature extraction modules along with several Dimensional Decomposition Residual (DDR) blocks with various dilations to effectively capture spatial structures and rich contextual information across varying receptive fields.
>
> Specifically, multi-scale feature extraction modules capture hierarchical contextual information by aggregating features across multiple receptive fields. It employs parallel branches of $3 \times 3 \times 3$ convolutions to replace $5 \times 5 \times 5$ and $7 \times 7 \times 7$ convolutions, which are progressively stacked and merged at multiple levels. Residual connections are integrated throughout the block to facilitate feature reuse and gradient flow.
>
> The Dimensional Decomposition Residual (DDR) blocks begin with a $1 \times 1 \times 1$ convolution for channel adjustment, followed by a series of three directional convolutions: $1 \times 1 \times 3$, $1 \times 3 \times 1$, and $3 \times 1 \times 1$, which capture contextual dependencies along the depth, height, and width dimensions, respectively. The outputs of these directional convolutions are then fused through an additional $1 \times 1 \times 1$ convolution. A residual connection is incorporated by adding the input feature map to the fused output.
>
> The Latent Mapping Network in Section 4.2 adopts the same architecture as the Encoder of the Variational Autoencoder (VAE). The VAE encoder consists of two down-sampling blocks, each comprising four consecutive convolutional layers. Every pair of convolutional layers is followed by a Batch Normalization layer and a ReLU activation function. Following these operations, a downsampling convolutional layer is applied, which is also followed by Batch Normalization and ReLU. We will add this architectural information in the camera-ready version and provide a corresponding figure to clearly depict the structure of the SSEN and Latent Mapping Network. We would also like to emphasize that we intend to make our code public to ensure a comprehensive understanding of our technique.
>
> > Section 4.4 introduces x(t), y(t), and h(t), but does not link them to z_TMB and f_TMB; a brief explanation and connection are needed.
>
> Thank you for the constructive comment. Advanced state space models (SSMs), such as the Structured State Space Sequence Model (S4) and Mamba, represent a class of systems that map a one-dimensional continuous input sequence $x(t) \in \mathbb{R}$ to an output $y(t) \in \mathbb{R}$ via a hidden state $h(t) \in \mathbb{R}^{N}$. These models are typically formulated using linear ordinary differential equations (ODEs), defined as:
>
> $h^{\prime}(t) = Ah(t) + Bx(t), \quad y(t) = Ch(t),$
>
> where $A\in \mathbb{R}^{N \times N}$ and$B\in \mathbb{R}^{N \times 1}$, $C\in  \mathbb{R}^{1 \times N}$ denote the state matrix, input matrix, and output matrix, respectively. Since deriving the analytical solution for $h(t)$ is often intractable and real-world data is typically discrete, we discretize the system as follows:
>
> $h(t) = \overline{A}h(t-1)  + \overline{B} x(t), \quad y(t) = Ch(t),$
>
> where  $\overline{A}= exp\left (\triangle A \right )$ and $\overline{B}=\left ( \triangle  A \right )^{-1} \left ( exp\left ( \triangle  A \right ) - I  \right ) \cdot \triangle B $ are the discretized state parameters and $\triangle$ is the discretization step size. In the context of our Triple Mamba layer, the input $x(t)$ corresponds to a residual-enhanced representation:
>
> $z_{TMB}(t) =LN(f_{TMB}(t))  + f_{TMB}(t),$
>
> where $f_{TMB}(t)$ is the raw input features prior to residual layer normalization $LN$. The temporal dynamics of the Mamba input are thus governed by:
>
> $h(t) = \overline{A}h(t-1)  + \overline{B} z_{TMB}(t), \quad y(t) = Ch(t),$
>
> To capture multi-directional dependencies, the Triple Mamba layer applies three separate Mamba modules, each operating on the same input $z_{TMB}(t)$ but with distinct ordering strategies: forward, backward, and random inter-slice directions. The output of the $i^{th}$ Triple Mamba layer is computed as:
>
> $\psi_{i}(z_{TMB}(t)) = \psi_{i}^{f}(z_{TMB}(t)) + \psi_{i}^{b}(z_{TMB}(t)) + \psi_{i}^{u}(z_{TMB}(t)),$
>
> where $\psi_{i}^{f}$, $\psi_{i}^{b}$, $\psi_{i}^{u}$ represent the outputs from the forward, backward, and random inter-slice directions, respectively. This formulation can be succinctly expressed as:
>
> $Triplemamba_{i}(z_{TMB}(t)) = Mamba_{i}^{f}(z_{TMB}(t)) + Mamba_{i}^{b}(z_{TMB}(t)) + Mamba_{i}^{u}(z_{TMB}(t)).$
>
> We will include these details in the camera-ready version (supplementary material if the space does not permit it in the main paper).
>
> > Lines 286–299 lack a clear reference to the corresponding table.
>
> Thank you for the comments. We will add the corresponding table number in the text.
>
> > The sketch and pseudolabels in Fig. 4 often appear to come from different, unrelated scenes.
>
> The difference is due to a temporal discrepancy between the data sources: the 3D ground truth annotations in SemanticKITTI were collected around 2013, whereas the satellite imagery used for Pseudo-labeled Satellite Image Annotations (PSA) was captured approximately in 2025. This temporal gap may lead to minor inconsistencies in spatial structures. Also, the PSA are generated through a weakly supervised pipeline that aligns satellite imagery with text embeddings, which enables scalable semantic supervision but can introduce semantic or geometric inaccuracies. In contrast, the sketches are derived directly from the 3D ground truth, ensuring temporal consistency with the underlying scenes. As a result, sketches serve as a reliable spatial reference that helps mitigate the domain gap introduced by the satellite imagery. We will explicitly note this in the camera-ready.
>
> > Consider citing recent methods like DynamicCity and Towards Generating Realistic 3D Semantic Training Data for completeness.
>
> We will cite and discuss DynamicCity [1] and 3DiSS [2] in the final version. These works are indeed relevant to our study. We sincerely thank the reviewer for mentioning them.
>
> [1] DynamicCity: Large-Scale 4D Occupancy Generation from Dynamic Scenes
>
> [2] Towards Generating Realistic 3D Semantic Training Data for Autonomous Driving.
>
> > Are 3D GT classes used to filter incorrect CLIP pseudolabels during generation?
>
> During pseudo-label generation, we do not directly filter the CLIP outputs using the 3D ground truth semantic classes. Instead, we utilize the set of semantic labels present in the 3D ground truth scenes as candidate classes for predicting labels in the satellite imagery. After the initial predictions are made, we conduct a manual validation step, where the predicted class distributions are compared against the bird's-eye view (BEV) projections of the 3D ground truth. This post-hoc verification ensures that the retained pseudo-labels are semantically consistent with the underlying scene context, thereby enhancing the accuracy and reliability of the annotation process.
>
> > Are the textual descriptions in Fig. 1 included in the dataset?
>
> Yes, we include textual descriptions used in Fig.1 in the dataset. It will be released to support further research.
>
> > The sketches are too precise to be human-drawn.
>
> We agree that human-drawn sketches are an important direction. Thanks for acknowledging our SketchSem3D dataset as a first step towards generating a 3D scene from human-drawn sketches. Based on our benchmark, future work will incorporate freehand or user-generated sketches to better approximate real-world input conditions and further enhance the model's generalization and practical usability.

---

> ### Comment · Reviewer_fZuf · 2025-08-05
>
> The authors have addressed and clarified my concerns in their rebuttal. Still, the mathematical notations and explanations in their rebuttal should be added to the main paper to enhance the clarity of the methodology section. I suggest adding it to the main paper, and in case space is needed, the authors can consider moving Figure 3 (the architecture) to the supplement. I see the mathematical explanation as more relevant to be in the main paper rather than the architecture figure. Therefore, I have raised my rating since the rebuttal clarified my concerns.

---

> > ### Author Response · Authors · 2025-08-05
> >
> > Dear Reviewer fZuf,
> >
> > Thank you for your positive feedback and the upgraded score! As suggested, we will incorporate the mathematical notations and explanations from the rebuttal in the final main paper to improve clarity. If needed, we will move Figure 3 to the supplementary material to ensure space for these additions.

---

### Official Review · Reviewer_4gZF · 2025-07-01

**Clarity:** 2
**Significance:** 3
**Originality:** 3
**Rating:** 4
**Confidence:** 4

**Summary:**

The paper introduces a dataset of outdoor scenes with paired sketches, semantic maps, and semantic voxels. It presents a pipeline that automatically constructs the dataset from existing semantic outdoor datasets, ground-truth semantic voxels and satellite imagery. The resulting dataset is approximately 2.7 times larger than existing BEV-conditioned datasets and supports more flexible conditioning. Furthermore, the authors propose an LDM model with Cylinder Mamba diffusion to preserve spatial relationships for conditioned 3D scene generation.

**Questions:**

1. Section 4 lacks sufficient details about key components. For the CSCB and DDR blocks, additional architectural tables or figures in the supplementary would have been helpful. The details regarding the cylinder mamba blocks is also quite limited. Such as the functionality and architectural details of the triple mamba mechanism as well as the ordering mechanism in the cylinder mamba block.

2. The diffusion model is conditioned on both PSA and sketches. However, the necessity of the sketch is unclear since the PSA alone should provide sufficient information for 3D scene generation. A more user friendly approach could involve a separate network to predict PSA from sketches. Finally, it's unclear to me why the pseudo labels from satellite images are needed as opposed to directly using the BEV projections of the semantic voxels.

3. The paper also lacks relevant conditioned scene generation baselines in table 2. As 3D latent diffusion and 3D DiT were not designed for 3D scene generation specifically. For example, the proposed method could have been run on NuScenes using BEV images as input to compare against UrbanDiff. This would also help understand whether the proposed task is easier or harder compared to BEV conditioned NuScenes. BlockFusion [1] is also another potential baseline that can be compared.

4. I found the 3D semantic scene completion results in table 4 to not be very informative. As the other methods takes monocular or stereo inputs, while the proposed method uses SK+PSA. SK+PSA may offer an easier setting as the semantic labels are directly provided to the model, whereas working with raw images is more challenging and requires stronger feature extractors. As a result, making direct comparisons with other methods less meaningful.

I think there is value in establishing unified datasets and benchmarks for evaluating 3D scene generation as well as offer more flexible inputs. If the above concerns can be addressed during the rebuttal I am willing to raise my rating.

[1] Wu, Zhennan, et al. "Blockfusion: Expandable 3d scene generation using latent tri-plane extrapolation." ACM Transactions on Graphics (TOG) 43.4 (2024): 1-17.

**Ethical Concerns:**

["NO or VERY MINOR ethics concerns only"]

**Final Justification:**

I have raised my rating as the authors have addressed my concerns regarding the experiments. However, the model section requires additional clarifications and details that the authors have promised during the rebuttal.

**Limitations:**

yes

**Quality:**

3

**Strengths And Weaknesses:**

## Strengths

The dataset is larger than previous BEV based NuScenes and provides more flexible conditioning. The description of the dataset construction is clear and the model also show better quantitative results with respect to the chosen baselines.

## Weaknesses

On the writing side, the clarity of the proposed method can be improved by giving more detail regarding the key components of the model. I also have concerns about the dataset, particularly the necessity of including both sketches and pseudo-labeled annotations as inputs. Additionally, the paper also lacks some more baselines that would make the evaluation more complete, and the validity of the semantic scene completion comparisons in section 5.2 is questionable. More details are given in the questions below.

---

> ### Author Rebuttal · Authors · 2025-07-30
>
> We sincerely thank Reviewer 4gZF for their comments and appreciation of our work, especially noting our dataset's size, flexibility, and strong baseline performance. In response to the concerns expressed in Weaknesses and Questions, we provide the following answers:
>
> > Supplementary architectural figures for CSCB and DDR blocks are needed. Details on the Cylinder Mamba, including the triple Mamba mechanism and its ordering process, are also insufficient.
>
> Thank you for the constructive comment. As shown in Figure 3, the Cylinder Mamba block integrates the Triple Mamba and C-Mamba layers to enhance representational capacity by combining Cartesian and cylindrical spatial representations. The Triple Mamba layer models feature dependencies along forward, backward, and inter-slice directions in Cartesian space. Although architecturally identical to the C-Mamba layer, the detailed structure of C-Mamba is explicitly illustrated in Figure 3 for clarity. The key distinction lies in the input domain: Triple Mamba operates on Cartesian grids, whereas C-Mamba processes data in cylindrical coordinates.
>
> In the C-Mamba layer, inputs are transformed into cylindrical coordinates and sorted by angular, radial, and vertical indices $(\theta, r, z)$ to align with the egocentric geometry of street-view scenes, enabling directional context modeling. After processing, features are mapped back to Cartesian space (sorted according to $(x, y, z)$) and fused with those from the Triple Mamba layer, allowing the model to jointly exploit radial and axis-aligned spatial cues.
>
> We will further improve clarity by adding architectural figures and more details on the triple Mamba mechanism and its ordering process in the final version. Please note, our source code will also be released to facilitate comprehensive understanding of our method.
>
> > The necessity of incorporating sketches remains unclear; PSA can be predicted from sketches. Why not directly utilize the BEV projections of semantic voxels?
>
> Thank you for the comment. Sketches offer clear and structured spatial layouts that effectively complement pseudo-labeled satellite annotations (PSA). Since PSA is generated by aligning recent satellite imagery with text prompts, it is susceptible to noise and spatial inconsistencies, particularly due to the temporal discrepancy between contemporary satellite data (e.g., from 2025) and the older LiDAR-based ground truth in SemanticKITTI (2013). Incorporating sketches mitigates these issues by providing reliable spatial priors, and jointly using both inputs enables more flexible and robust conditioning, thereby enhancing spatial accuracy.
>
> We conducted an ablation study by training CymbaDiff using only PSA inputs, which required code adjustments and excluded sketch cues. As shown in the table below, the resulting performance drop underscores the essential role of sketch guidance in enhancing 3D scene generation quality and fidelity.
>
> | Input Modality       | Dataset        | 3D FID ↓ | 3D MMD ↓ |
> |----------------------|----------------|----------|----------|
> | PSA only   (Ours)          | SemanticKITTI  | 83.31    | 0.06     |
> | PSA + Sketch (Ours)  | SemanticKITTI  | 40.67    | 0.04     |
> | PSA only     (Ours)        | KITTI-360      | 155.62   | 0.08     |
> | PSA + Sketch (Ours)  | KITTI-360      | 107.53   | 0.08     |
>
> Predicting PSA from sketches remains an underexplored direction, which we plan to investigate in future work using the annotations provided in our benchmark. Notably, we deliberately refrain from using BEV projections of semantic voxels, as dense 3D annotations are often unavailable in practical settings. In contrast, satellite imagery serves as a scalable and accessible source of weak supervision, making our approach more broadly applicable under realistic data constraints.
>
> > The method could be tested on BEV-based NuScenes to compare with UrbanDiff and assess task difficulty, and provide BlockFusion results on SketchSem3D.
>
> We selected 3D Latent Diffusion and 3D-DiT as baselines due to the limited availability of open-source 3D scene generation models. Although not originally designed for semantic voxel generation, both models have shown strong performance in image and 3D object synthesis, and their general architectures can be effectively adapted to the SketchSem3D dataset to establish strong baselines.
>
> The pre-processed BEV-conditioned NuScenes dataset used in UrbanDiff is not publicly available, so we cannot train the proposed CymbaDiff on this dataset. To address the reviewer's concern, we directly applied our model, trained solely on the Sketch-based SemanticKITTI dataset, to the Sketch-based nuScenes validation dataset without any fine-tuning. Evaluation on the nine overlapping semantic classes shared between the two datasets shows that our CymbaDiff achieves a 3D FID of 183.37 and a 3D MMD of 0.09, outperforming the popular UrbanDiff model on the BEV-conditioned NuScenes dataset that achieves 3D FID and 3D MMD scores of 291.4 and 0.11 (lower is better). In addition, the proposed SketchSem3D dataset offers a more practical alternative to BEV-conditioned NuScenes. We will make this dataset publicly available to facilitate future research.
>
> BlockFusion serves as a strong baseline for 3D scene generation. However, since it was originally designed as a 2D tri-plane denoiser, adapting it to our 3D latent space led to unstable training and suboptimal performance, with the loss failing to converge.
>
> > Table 4 is less informative, as SK+PSA offers an easier setting than raw-image-based methods.
>
> Our work explores a new research direction, and currently, there are no directly comparable methods using the same input modalities. For this reason, we compare CymbaDiff with existing methods that use monocular or stereo RGB inputs, which are common in semantic scene completion. These comparisons highlight the differences, gaps, and challenges between input modalities, as our approach is based on sketches (SK) and pseudo-labeled satellite image annotations
> (PSA) rather than RGB images. While stereo images can provide geometric depth cues for 3D reconstruction, our results demonstrate that SK and PSA inputs offer a flexible and effective alternative, particularly in scenarios where conventional RGB inputs are unavailable or impractical, such as remote sensing. In the final version, we will provide further explanation to clarify these distinctions and prevent possible misinterpretation.
>
> We appreciate the reviewer mentioning their willingness to raise the rating, and hope that the above responses help in that regard. We look forward to discuss further in the discussion phase, if needed.

---

> > ### Comment · Reviewer_4gZF · 2025-08-04
> >
> > Regarding the additional experiments, the authors have addressed my concerns. After reading the other reviews, I also noted other reviewers have also raised issues about the clarity of the model section. The model description remains a major weakness of the paper. And it is essential that the authors incorporate the additional details described in the rebuttal with formal mathematical rigor for parts of the model (see Reviewer fZuf) for the paper to be complete. Nevertheless, I will raise my rating as my concerns with the experiments have been addressed.

---

> > > ### Author Response · Authors · 2025-08-05
> > >
> > > Dear Reviewer 4gZF,
> > >
> > > Thank you for your positive feedback! We are glad that our rebuttal has addressed your concerns. Your constructive comments have been very helpful in refining our work, and we will incorporate the additional details as described in the rebuttal in the final paper.

---

### Note · Authors · 2025-08-13

We sincerely thank the reviewers and AC for their valuable time and constructive feedback.

This submission introduces **the first large-scale 3D outdoor semantic scene generation model and benchmark (2.7 times larger than the existing BEV-based NuScenes), built from freehand-like sketches and pseudo-labels**, considerably reducing manual annotation requirements and enabling efficient training data generation for applications such as urban-scale simulation and autonomous driving. It demonstrates state-of-the-art (SOTA) performance for both semantic scene generation and completion, while enabling a much more flexible supervisory signal.

After discussion, ***reviewers 4gZF, fZuf, Ws1Y explicitly noted that their concerns have been addressed***, and gave some recommendations on improving clarity. We will follow their recommendations to improve the clarity in the camera-ready paper and will also make our complete code, dataset and trained models public for reproducibility of results. We are grateful to the reviewers for their precise recommendations on how to improve clarity, which can be conveniently incorporated in the camera-ready.

During discussion, ***reviewer pTZW also acknowledged that many of their concerns were well addressed*** by our initial responses, and asked two further questions. We clarified both, however it seems that the reviewer could not respond further due to limited time. We are confident that we have well addressed the comments of pTZW and hope that they agree with our responses.

We emphasize that our submission introduces a first-of-its-kind method for outdoor 3D voxel scene generation with freehand-like sketches and pseudo-labels conditioning. Despite using a far more flexible and efficient supervisory signal, our method outperforms SOTA scene generation and completion techniques that require precise RGB and stereo input (often not available in practice).

Thank you.

Kind regards,

Authors.

---

### Decision · Program_Chairs · 2025-09-17

**Decision:**

Accept (poster)

**Comment:**

The AC and the reviewers thank the authors for their response.
This paper proposes the first benchmark for outdoor 3D semantic scene generation, SketchSem3D, and designs the CymbaDiff framework to achieve higher spatial consistency and semantic fidelity, advancing research in this field.

After the rebuttal, one reviewer leaned toward acceptance, and two reviewers leaned toward borderline acceptance. However, another reviewer expressed concerns about generating realistic scenes, downstream applications, and the overall quality, thus giving a borderline rejection.

After carefully reading the paper, the reviewers’ comments, and the authors’ response, the AC acknowledges the contribution of establishing the first large-scale outdoor 3D semantic scene generation model and benchmark. The proposed large-scale benchmark SketchSem3D and the CymbaDiff method demonstrate promising direction and performance. Moreover, the construction process based on abstract freehand sketches and pseudo-labels significantly reduces the need for manual annotations. These aspects provide new perspectives for research in this direction.

The AC believes that after addressing the reviewers’ comments and fixing minor issues, the paper is suitable for acceptance.
The AC recommends acceptance.